ature
# ROYAL SOCIETY
# OPEN SCIENCE

environmental science/behaviour

bycatch, null model, permutation, social network analysis, sea turtle

**Author for correspondence:**
William N. S. Arlidge
e-mail: william.arlidge@igb-berlin.de

# Assessing information-sharing networks within small-scale fisheries and the implications for conservation interventions

William N. S. Arlidge[1,3,4], Josh A. Firth[2,5],
Joanna Alfaro-Shigueto[6,7,8], Bruno Ibanez-Erquiaga[9,10],
Jeffrey C. Mangel[6,7], Dale Squires[11] and
E. J. Milner-Gulland[1]

[1]Interdisciplinary Centre for Conservation Science, Department of Zoology, and
[2]Edward Grey Institute, Department of Zoology, University of Oxford, Oxford OX1 3PS, UK
[3]Department of Biology and Ecology of Fishes, Leibniz-Institute of Freshwater Ecology and Inland Fisheries, Müggelseedamm 310, 12587 Berlin, Germany
[4]Faculty of Life Sciences, Humboldt-Universität zu Berlin, Invalidenstrasse 42, 10115 Berlin, Germany
[5]Merton College, University of Oxford, Oxford, UK
[6]ProDelphinus, Calle José Galvez 780-E, Lima 15074, Perú
[7]School of Biosciences, University of Exeter, Cornwall Campus, Penryn, Cornwall TR10 9FE, UK
[8]Carrera de Biologia Marina, Universidad Científica del Sur, Lima, Perú
[9]Section for Coastal Ecology, Technical University of Denmark, National Institute of Aquatic Resources (DTU Aqua), Lyngby 2800, Denmark
[10]Asociación CONSERVACCION, Lima, Peru
[11]NOAA Fisheries Southwest Fisheries Science Center, La Jolla, CA, USA

WNSA, 0000-0002-1807-4150; JAF, 0000-0001-7183-4115;
BI, 0000-0002-9954-5953; JCM, 0000-0002-9371-8606;
EJM, 0000-0003-0324-2710

The effectiveness of behavioural interventions in conservation often depends on local resource users' underlying social interactions. However, it remains unclear to what extent differences in related topics of information shared between resource users can alter network structure—holding implications for information flows and the spread of behaviours. Here, we explore the differences in nine subtopics of fishing information related to the planned expansion of a community co-management scheme aiming to reduce sea turtle bycatch at a small-scale fishery in Peru. We show that the general network structure detailing information sharing about sea turtle bycatch is dissimilar from other fishing information sharing. Specifically, no significant degree assortativity (degree

homophily) was identified, and the variance in node eccentricity was lower than expected under our null models. We also demonstrate that patterns of information sharing between fishers related to sea turtle bycatch are more similar to information sharing about fishing regulations, and vessel technology and maintenance, than to information sharing about weather, fishing activity, finances and crew management. Our findings highlight the importance of assessing information-sharing networks in contexts directly relevant to the desired intervention and demonstrate the identification of social contexts that might be more or less appropriate for information sharing related to planned conservation actions.

# 1. Introduction

Managing the use of common-pool natural resources such as fishes often involves behaviour-change interventions with resource harvesters [1,2]. These can include interventions like the enforcement of rules, the adoption of new technologies, social marketing and education campaigns [3–5]. Informed by other behavioural-change disciplines like public health and social marketing, biodiversity conservation researchers and practitioners are increasingly interested in the social structures of communities targeted for interventions to predict how information sharing can enable pro-environmental behaviours to spread beyond a target group [6,7]. Understanding how the structures of information-sharing networks vary with the topic of information shared, therefore, has important implications for designing successful behavioural-change interventions in conservation.

Throughout the world's fisheries, bycatch (defined here as incidental catch that is either 'unused or unmanaged' [8]) remains a critical issue for marine species, ocean ecosystems and fishing communities [9–11]. Bycatch is notably problematic for taxonomic groups that are either highly migratory or that have conservative life-history characteristics including sea turtles, seabirds, marine mammals, elasmobranchs and corals [12]. What is more, managing bycatch is a particularly intractable issue among geographically dispersed populations of resource-constrained small-scale fishers in low- and middle-income countries [13,14]. Small-scale fisheries are hugely important to many coastal communities, employing more than 90% of the world's wild capture fishers and fish workers [15]. Yet, the bycatch issues in small-scale fisheries remain widespread and under-reported [16–18].

Small-scale fisheries often lack institutional capacity and have weak state oversight [19]. In such instances, individual decision-makers are subject to fewer legal constraints and are more prone to influence by their peers [20]. For example, Alexander *et al*. [21] found that fishing experience dictates the influence among small-scale fishers in Jamaica, with older fishers and information brokers having discrete roles in shaping catch patterns for large- and small-sized fish species, respectively. The adoption of pro-environmental behaviours in small-scale fisheries often occurs through social influence [1,22,23] and social reinforcement [24], which result from interpersonal communication, and the evaluation of credibility and social norms between peers [25–28]. In particular, social network analysis has proven useful for understanding the social dynamics of information-sharing between fishers [29], considering the establishment of common rules and norms among stakeholders [30,31] and understanding complex social–ecological interactions to enhance conflict resolution strategies [32].

When empirically exploring peer-to-peer information exchange in fishing communities, it can seem intuitive to build information-sharing networks by asking fishers with whom they exchange information about fishing [33–35]. Yet, individuals sharing one topic of information may not be the most central individuals when sharing other closely related information topics (following a similar logic to topic limited opinion leaders in online social networks [36,37]). Would, for example, a bycatch-specific information-sharing network be more informative for transmitting information about the existence and aims of a bycatch reduction strategy over other topics of fishing information of relevance to the intervention in question? Here, we explore the assumption that the structure of the network (i.e. which fishers are socially tied to one another, and who may share information) is consistent across different information-sharing networks that relate to a planned conservation intervention. This assumption implies that the social ties measured for one topic of information will also be important for spreading the conservation information of interest for another closely related information topic. Indeed, if individuals' social behaviour remains consistent across different aspects of information sharing about fishing, in terms of which individuals they form social relationships with and the number of relationships they form, then the social networks across these contexts are expected to be correlated [38,39]. As individuals who share information to a particular topic, they may be more likely than a non-

connected pair of individuals (dyad) to share a different topic of information (i.e. two fishers who know each other versus two that do not know each other). We, therefore, hypothesized that information-sharing networks across multiple topics of information that relate to fishing would be correlated. Yet, specific networks may be strongly correlated to one another, while other networks may be less correlated.

We focus on a coastal fishing community in Peru with problematic sea turtle bycatch [16,40–42]. At the study site, a local not-for-profit is trialling a community co-management scheme aiming to reduce sea turtle bycatch [43]. This initiative intends to create direct incentives for sea turtle bycatch reduction by giving price premiums to fish caught by fishers that follow best-practice bycatch reduction guidelines such as using light-emitting diodes on nets [44]. Timely bycatch information is conveyed to fishers by the not-for-profit [45], which intends to expand the community co-management scheme, first to more fishers within the target community, and second to similar communities along Peru's coast. The community co-management expansion could be more cost-efficient if resource managers better understand how messages about the sea turtle bycatch reduction initiative's existence and aims might spread.

A structural comparison of multiple fishing information-sharing networks will allow us to explore whether it would be possible to design an effective (if sub-optimal) sea turtle bycatch intervention by identifying and targeting influential individuals in a network sharing information on other topics related to the intervention. This is a pertinent question because information-sharing about sea turtle bycatch might be sensitive and therefore hard to quantify, or it may be that the information-sharing networks for other topics are already known so the cost of describing a sea turtle bycatch-relevant network would not need to be incurred. It is also interesting more generally to explore how the networks for sharing different types of fisheries information resemble each other, as this may give insights into how different kinds of information are perceived by fishers.

In this study, we apply a social network analysis and permutation-based null model approach to assess whether networks of small-scale fisher's information sharing about sea turtle bycatch are structurally similar to other topics of fishing information-sharing networks (table 1). We test the assumption made by conservation researchers and practitioners that knowledge about peer-to-peer information-sharing networks should be transferable to a related information-sharing network of interest (other fishing issues and sea turtle bycatch, in our case). We provide insight into comparing information-sharing networks within a social system of high conservation interest. Finally, we conclude by discussing how our findings can contribute to understanding how information related to conservation interventions may spread socially.

# 2. Material and methods

## 2.1. Study system

During our survey period of 1 July–30 September 2017, San Jose, Lambayeque, Peru (6°46′ S, 79°58′ W) was home to 168 small-scale commercial gillnet skippers that fish throughout the year. We surveyed 165 fishers representing 98.2% of the gillnet skippers at the site (electronic supplementary material, figure S2b and table S1). Gillnet skippers in San Jose are known to capture sea turtles in high numbers [16,40,46]. Green turtles (*Chelonia mydas*) are captured most frequently, followed by olive ridley turtles (*Lepidochelys olivacea*) and leatherback turtles (*Dermochelys coriacea*) [43]. At the time of the study, five gillnet skippers and their crew were involved in a trial community co-management scheme operating from San Jose that requires fishers to use light-emitting diodes on their nets to reduce sea turtle bycatch [44]. Skippers were deemed active if they fished from the San Jose port with gillnets in the winter of 1 July–30 September 2017. The social network was surveyed during winter as skippers actively fishing during these months are established fishers in the San Jose community throughout the year. We define gillnets as encompassing surface drift gillnets and fixed bottom gillnets in single or trammel net configurations. The total San Jose gillnet skipper population ($n = 168$) was determined using a combination of membership lists of the two main fishing groups in San Jose, lists of boats towed in and out of the water with tractors, and key informant interviews (electronic supplementary material, Information).

## 2.2. Data collection

Detailed social network data were collected using a structured questionnaire with a fixed choice survey design. Respondents were asked to consider up to 10 individuals with whom they exchange useful information about fishing and whom they considered valuable to their fishing success. We classified

**Table 1.** Fishing information-sharing networks that relate to the intervention aiming to reduce sea turtle bycatch in San Jose, Lambayeque, Peru.

| full name | short name | description | broad categorization |
|---|---|---|---|
| sea turtle bycatch | T.Bycatch | sea turtle bycatch encounters including live releases and mortalities in nets | process of fishing, business and governance of fishing |
| gillnet type & maintenance | gear | changes made to net configuration (shifting rigging configurations from surface drift net to mid-water drift net or bottom-set net), and net maintenance | process of fishing |
| weather conditions | weather | ocean and weather conditions (e.g. wind, swell) | |
| fish location & catch sites | location | where fish might be located and where they have been travelling to fish | |
| fishing activity | activity | how many people fishing, who is fishing, who caught what | |
| vessel technology & maintenance | tech | existing and new technologies used onboard the vessel (e.g. echo sounder, compass) and vessel maintenance (e.g. hull repairs, painting) | |
| fishing regulations | regs | fishery policy and legislation | business and governance of fishing |
| fishing finances | finance | market prices, loans, fines, penalties | |
| crew management | crew | the hiring and instructing of crew onboard the vessel | |

nine subtopics of fishing information that are of relevance to the community co-management scheme aiming to reduce sea turtle bycatch at the study system and which we expect gillnet skippers to exchange (table 1). As each nominee was given by the respondents, they were asked to highlight which topic of fishing information they discussed with each nominee. For each topic of fishing information, respondents were asked to consider relationships that they have had with other skippers, vessel owners, crew members, other fishery leaders, fishery management officials, members of the scientific community, boat launching/landing support, fish sellers/market operators, family members and any other stakeholders they fished or shared information with about fishing. Respondents were not asked who they receive information from. Interviews were undertaken verbally and respondents were not shown the questionnaire where responses were written (electronic supplementary material, Information). Questionnaires were trialled with fishers ($n = 8$) in the Santa Rosa fishing community 17 km down the coast from San Jose (figure 2a). Pilot study data were not included in this study's analysis. Fishers were interviewed in their native language (Spanish).

## 2.3. Statistics and reproducibility

### 2.3.1. Social network construction

A social network was created for nine subtopics of fishing information of relevance to the intervention aiming to reduce sea turtle bycatch in our study system (table 1). In each network, the nodes were the fishers, and the binary directed edges were the nominations by one fisher (sender) of another fisher (receiver) for this information type. All analysis was carried out in R [47] using the igraph package [48] for visualizing and processing the analysis and carrying out the network comparisons using the null models.

### 2.3.2. Structural differences across information-sharing networks

To investigate whether networks of information-sharing between individuals were similar across different subtopics, we examined the networks' structural properties in terms of their degree assortativity and the variance and mean of individual centrality (table 2). To account for the effect of basic characteristics of the networks (e.g. number of ties, degree distributions), we compared these observed summary statistics with null models, which allowed inference of structural differences and similarities over-and-above that expected from these simple differences using null models (figure 1).

Network null models (routines that generate different types of null datasets against which the observed dataset can be compared) are a group of statistical models commonly applied in network analysis. Specifically, null models are especially useful when investigating hypotheses in datasets, control groups are difficult to establish, exogenous treatments are unavailable, and observations may be missing or biased [55–57]. As such, null model methods are important because network data comprise non-independent observations of multiple individuals, and small variations in how data are collected between respondents can easily generate patterns that appear as social structure [57,58]. Null models have been applied to network data in sociology since the 1970s [55] and discipline-specific developments have subsequently been made to statistical models such as exponential random graph models [59,60], conditional uniform graph tests [61–63] and quadratic assignment procedure tests [64–66]. Since the mid-1990s, the field of ecology has also made extensive use of null models to develop specialized hypothesis testing routines and treat underlying uncertainty or data collection methodology biases when interrogating non-human animal network data [67–69]. Here, we expand the application of the permutation-based null model approach routinely used in ecology to human social networks, and which has also been applied in the field of epidemiology for assessing human contact tracing disease control measures, to a human information-sharing social network [70].

### 2.3.3. Degree assortativity

The assortativity coefficient (akin to homophily [51]) measures the extent to which central fishers are connected to other central fishers, and peripheral fishers are connected to other peripheral fishers based on a particular trait [49,50,71]. The level of degree assortativity in a network is known to have important social implications for information transfer, and for the operation and emergence of competition and cooperation [50,51]. Degree assortativity can, for example, influence the potential for a social contagion to spread, given its starting point [25,72]. To inform the planned expansion of the sea turtle bycatch reduction initiative in our study system, we were interested in understanding the general structure of multiple subtopics of fishing-related information that relate to the intervention and how the information-sharing networks relate to one another. Moreover, evaluating who talks to whom (i.e. directed network ties) has implications for how information may or may not flow. This is because individuals within a network can be highly central (generally nominated by many others) but just receive information—resulting in knowledge accumulation and the impeding rather than facilitation of information flow [26,73]. Therefore, degree assortativity was the primary assortativity measure of interest as degree provides a measure of which fishers provide information to others (in-degree) and receive information from others (out-degree).

A degree assortativity coefficient of zero represents randomness. Positive values demonstrate degree assortativity in which high-degree nodes tend to connect to other high-degree nodes, whereby a score of 1 would indicate that the network is assorted by individuals' degree to the maximum extent. Negative values represent disassortment (i.e. high-degree individuals are more likely to be associated with low degree individuals). When fishers of similar centrality are disassorted in a community, those networks do not always score -1 because the minimum value depends on the number of fishers and the relative number of ties within each group [50]. For each of the information-sharing networks, we first calculated the assortativity by in-degree (the number of nominations each interviewed fisher received from their peers in term of discussing a particular subtopic of fishing information). However, as fishers differed in the number of nominations they made for each information-sharing topic, we also calculated the assortativity by out-degree (the number of nominations each fisher made) to examine whether fishers were also disproportionately connected to others who make a similar number of nominations as themselves. As social networks often show assortativity by degree, we predicted that all the information-sharing networks would be positively homophilous by nominations made and nominations received (i.e. highly nominating and nominated fishers would be closely associated with

**Table 2.** Network metrics used to assess information-sharing network structure. Fishers (circles) and ties (arrows) outline the represented metric in the network. Red circles and arrows highlight the relevant structure of the network each metric measures. Black and white circles and black arrows highlight structures of the network that are not relevant to the metric measure in question.

| metric | network structure | description | theoretical use in conservation-relevant systems | example |
|---|---|---|---|---|
| degree assortativity |  | a preference for individuals to associate with others that are similar in degree (e.g. high in-degree) [49,50]. Akin to degree homophily [51]. | identifies individuals and pathways of individuals that could facilitate widespread diffusion of information about conservation initiatives in a community of conservation interest | a comprehensive, socio-centric network study of the Hadza hunter–gatherers of Tanzan was undertaken. Hadza networks were positively assorted by degree. People with higher in-degree named more social contacts, and people with higher out-degree were more likely to be named, even in models with controls. In other words, individuals who nominate more friends are more popular even among those they themselves did not nominate [52]. |
| node eccentricity |  | the furthest network distance between an individual and all other individuals in the network [53]. Equivalent to the inverse of some definitions of 'node closeness'. | can inform whether or not information relevant to a conservation initiative is shared in an even or clustered manner throughout a community on interest. This can inform how social norms and personal beliefs might affect information flow, which in turn can allow for conservation practitioners to tailor interventions to particular perspectives about a harmful activity (e.g. bycatch). | using social network analysis and several centrality measures including 'node closeness' (also equivalent to the inverse of some definitions of 'node eccentricity') the authors assess the structural nature and expanse of climate-based communication between professionals across sectors in the Pacific Islands region. Their results show a simultaneously diffuse and strongly connected network, with no isolated spatial or sectoral groups. The most central network members were shown to be those with a strong networking component to their professions [54]. |

highly nominating and nominated fishers, while peripheral fishers would be more likely to be connected to other peripheral fishers).

### 2.3.4. Eccentricity

As well as assortativity-based metrics, the variance in node centrality provides an informative and intuitive network measure regarding the uniformity of a network's structure, its resilience to perturbations and the influence of start-points on social contagions [74–76]. For this purpose, we used node eccentricity (igraph package [48]), which measures how far a fisher is from the furthest other in the network [53]. Node eccentricity can be particularly informative when investigating the flow of information and transmission of behaviours across a network following an intervention (table 2). Although this metric describes a fisher's position within the fishing community, the range of potential values it can take is not overly affected by permutations of the network structure in comparison with other more vulnerable metrics (e.g. betweenness, clustering coefficient) which are innately dependent on multiple aspects of the set structure of the network and are intuitively expected to differ largely from permutations by default. Finally, this metric is also relatively fast to compute; this is particularly useful when calculating it for many iterations of null networks. As such, we computed the variation in eccentricity in 'received nominations' (in-eccentricity) for each of the information-sharing networks.

### 2.3.5. Null models for structural differences

Drawing comparisons of network structure, correlations and fisher positions across different networks requires particular consideration because the general structure of the network (such as the number of ties or degree distributions) has a large effect on the observed values obtained from standard summary statistics. This structure can be taken into consideration by comparing networks with null permutations (controlled randomizations) of themselves and recalculating the same summary statistics on the null networks. Through comparing the observed values of the summary statistics with the distribution of those statistics generated from the null networks, insight can be gained into the actual differences between observed networks across other networks, over-and-above what is expected from simple properties such as the number of ties.

When calculating summary statistics (in-/out-degree assortativity, eccentricity) of each of the information-sharing networks, we also compared these with the values generated from permuting each of the networks separately. Specifically, we carried out edge permutations. The first edge permutation simply allowed the randomization of all incoming ties, while maintaining the number of nominations (outgoing ties) each individual made within this information-sharing network (termed edge null model 1—figure 1a). The second edge permutation was a more conservative version of this, allowing swaps of ties (which individuals nominated which other individuals in this information-sharing network) but maintaining the number of nominations each individual made in this information-sharing network (termed edge null model 2—figure 1b). Separately, for each of the information-sharing networks, 1000 permuted networks (of both of these permutation types) were generated and the distribution of the summary statistics was calculated for them.

### 2.3.6. Cross-network correlations

To reveal the extent to which the sea turtle bycatch information-sharing network can be correlated with the other networks evaluated, we examined the dyadic similarity between the different information-sharing networks. We used cross-network null models to compare the expected correlation between each network and subsequently determined how the observed correlation between each network was driven by fine-scale structure over-and-above that expected from the system's general social structure. While various metrics are available for considering similarities between networks [77,78], we chose to examine the relationship between each network of dyadic information-sharing nominations by calculating the correlation between the dyadic nominations on the network matrices. This approach is somewhat analogous to the Mantel test [79] (that tests the correlation between two matrices), yet as the networks were directed (and non-symmetrical), this was applied to the entire matrix rather than the lower triangle part (but excluding the diagonals because 'self-nominations' were not possible). The calculated correlation statistic represented the similarity/dissimilarity in the directed dyadic nominations among networks (who nominates whom), and these were compared with the distribution of the correlation statistic generated from the null models. To infer the extent to which

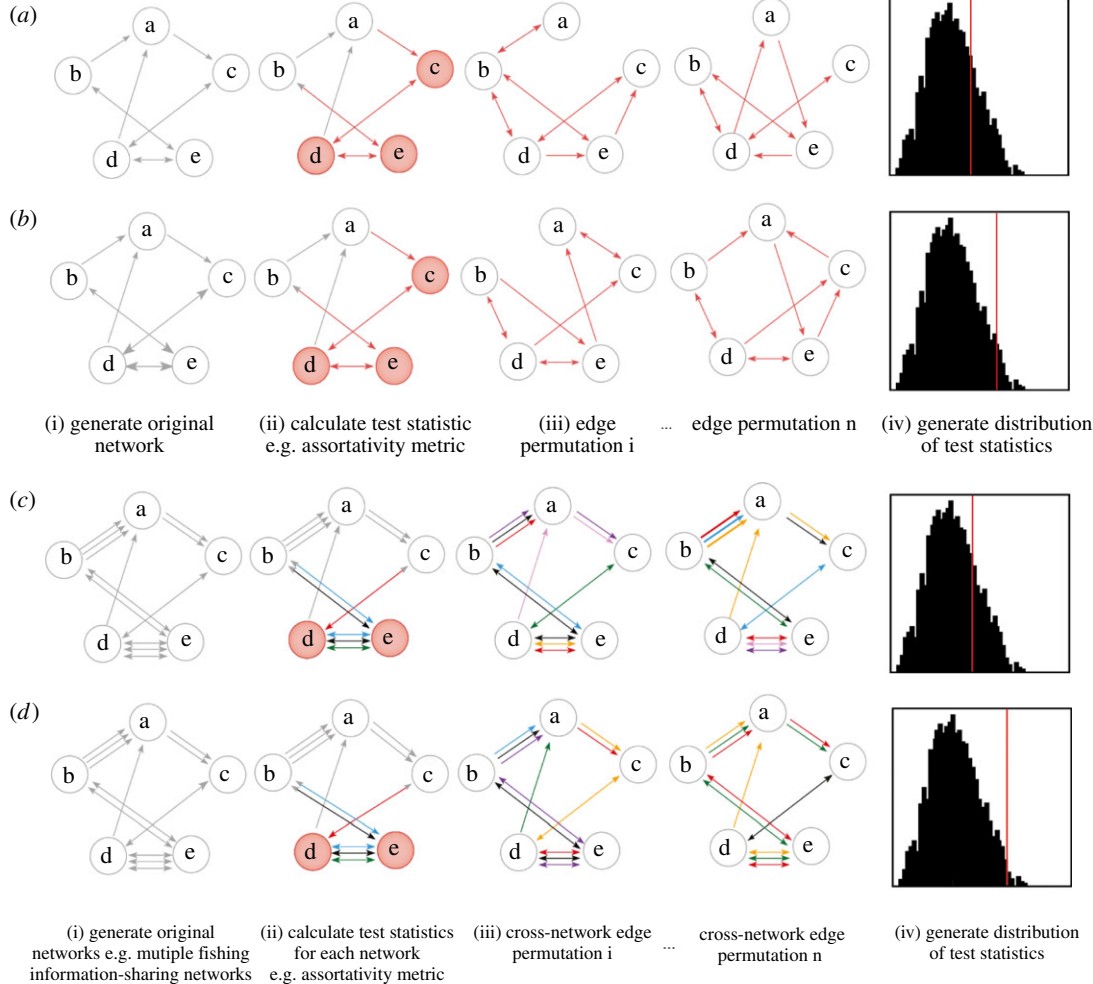

**Figure 1.** Schematic representation of edge-based permutation models with directed network data. Four main null model steps include (i) creating a social network from the observed data, (ii) calculating a test statistic, for example, a network-level metric like degree assortativity (high-degree fishers that are coloured red primarily connect to other high-degree fishers), (iii) randomizing the observation data (typically with 1000 permutations) and (iv) recording the distribution of possible test statistics. Conclusions can then by drawn by comparing the observed test statistics with the distribution test statistics, and the *p*-value calculated. Throughout the edge permutations, the fisher positions remain the same, but the configuration of edges between fishers change based on select criteria. The four null model examples shown are all used in this paper's analysis. (*a*) Outgoing edge permutation allows the randomization of all incoming links, while maintaining the number of nominations (outgoing links) each individual made, (*b*) edge permutation only allows the swap of links, by maintaining the number of nominees (incoming links) and nominations (outgoing links) each individual made in this information-sharing network. (*c*) Network swap permutation maintains each dyadic nomination, but randomizes the networks that these nominations were made in (i.e. when individual X nominated individual Y for information sharing within three different information-sharing networks (represented by different coloured arrows), the cross-network permutation allows these three nominations to be reassigned to any of the nine possible networks), and (*d*) conservative network swap permutation maintains each dyadic nomination, but randomizes the networks that these nominations were made in, while also controlling for the number of nominations that took place overall within each network (i.e. when individual X nominated individual Y for information sharing within three different information-sharing networks, these three nominations were reassigned among the networks in a way that was equal to the number of nominations in each network).

networks are more or less similar than expected under the general dyadic social structure, we carried out a cross-network null model: for each dyadic nomination across any of the networks, we randomized the networks that these nominations were made within (termed 'cross-network null model 1'—figure 1*c*). As an even more conservative version of a cross-network null model, we created a new version of these permutations and controlled for the number of nominations that took place overall within each network (termed cross-network null model 2—figure 1*d*; electronic supplementary material, figures S7 and S8).

# 3. Results

We constructed nine full fishing information-sharing networks. Of the 165 skippers surveyed, 151 nominated at least one gillnet skipper from the site as a key contact they talk to about fishing success, while 116 fishers from the site were nominated at least once by other fishers surveyed. The networks resulted in a total of 427 fisher-to-fisher nominations (i.e. ties between the 165 skippers interviewed) for one network or more (electronic supplementary material, table S1). On average, fishers had 2.8 fisher-to-fisher contacts with whom they had formed communication ties specific to fishing. Information-sharing networks per nomination averaged 7.7 (range 1–9). Fishers received on average 3.7 ties (range 1–15) for one or more information-sharing network. Across the nine information-sharing networks evaluated (table 1), sea turtle bycatch was discussed by fishers the least (61.6% of possible fisher–fisher ties). By contrast, fishing location and fishing activity were discussed by fishers most frequently (both in 97.9% of the possible fisher-to-fisher ties; electronic supplementary material, table S1).

## 3.1. Structural differences between information-sharing networks

We separately assessed degree assortativity (akin to degree homophily) and node eccentricity of the sea turtle bycatch information-sharing networks and each of the other networks of information sharing related to fishing (table 2). Across these networks, we compared how the observed statistics differed from edge-permutated versions of themselves. We considered the observed statistic to be significantly different from that expected under the null models when it fell outside the 95% range of the distribution of the statistics generated by the permutations (i.e. equivalent to significantly different at $p < 0.05$ level in a two-tailed test).

### 3.1.1. Degree assortativity

For each subtopic of fishing information (table 1), we evaluated degree assortativity (the propensity for a fisher to be connected to others who are similarly (dis-)connected; referred to as degree homophily in the social sciences), as this is a primary structural component of the network [49,50] (table 2). We found that networks of sea turtle bycatch information-sharing nominations show no significant degree assortativity in comparison with the edge permutation null models (observed stat: 0.038, edge null model 1: mean ± s.d. = −0.005 ± 0.059, $p = 0.512$; edge null model 2: mean ± s.d. = −0.011 ± 0.059; $p = 0.39$). As such, there was no evidence for a non-random tendency for highly nominated fishers to be disproportionately connected to other highly nominated fishers, nor for rarely nominated fishers to be disproportionately connected to other rarely nominated fishers. The sea turtle bycatch information-sharing network differed markedly in this regard from all of the other information-sharing networks' (figure 2c), all of which had significantly higher degree assortativity scores than expected from edge permutation null model 1. In addition, all the other information-sharing networks had significantly higher degree assortativity scores than expected from edge permutation null model 2 apart from the 'weather' and 'technology' networks, which fell outside the top 5% of the null network degree assortativity coefficients but were not significantly different in the two-tailed test (edge permutation model 2 two-tailed $p = 0.06$) (figure 2d).

### 3.1.2. Eccentricity

We found that sharing of information regarding sea turtle bycatch had a significantly lower variance in node eccentricity than expected under the null models controlling for simple properties such as the number of nominations and degree distributions (observed stat: 14.71, edge null model 1: mean ± s.d. = 41 ± 13.5, $p < 0.01$; edge null model 2: mean ± s.d. = 22.66 ± 5.335; $p < 0.05$). Importantly, sea turtle bycatch information sharing was again unique in this sense (figure 2d), as none of the other information-sharing networks was significantly lower than expected under null permutations of themselves (electronic supplementary material, table S2). Six of the eight other networks showed significantly higher variance in node eccentricity than expected from a null model of their structure, which illustrates a particularly stark contrast from the sea turtle bycatch information-sharing network. These results demonstrate less variation in individuals' centralities across the gillnet skippers than expected in terms of sea turtle bycatch information sharing. In other words, gillnet skippers are more similar in how they share information about sea turtle

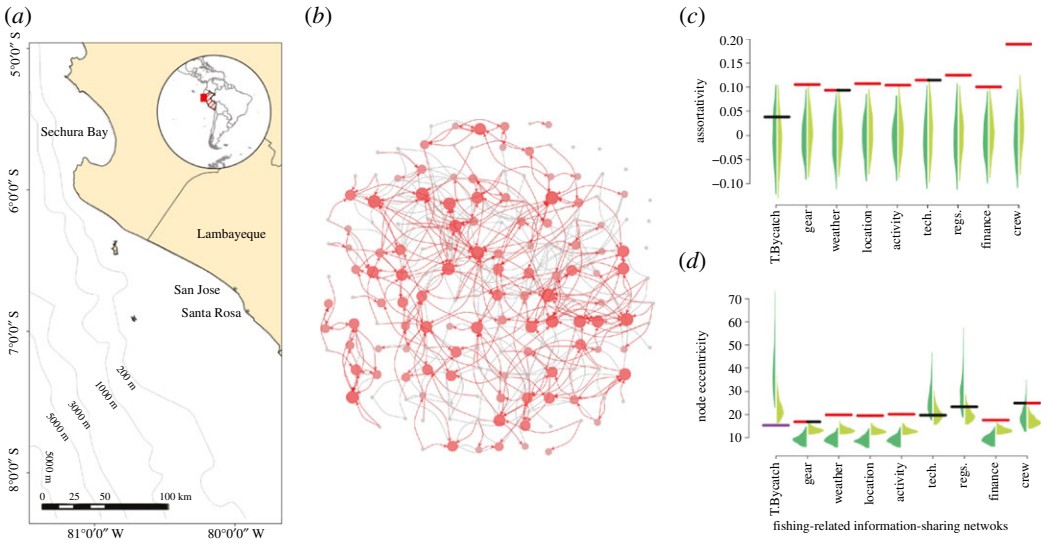

**Figure 2.** Structure of information-sharing in relation to sea turtle bycatch. (*a*) A map of the study site, San Jose, Lambayeque, Peru (6°46′ S 79°58′ W) and the surrounding coastline. (*b*) Illustrative network of the structure of information-sharing in relation to sea turtle bycatch. The nodes show each of the fishers surveyed, the adjoining lines show which fisher pairs (dyads) shared information in at least one information-sharing network, and nominations within the sea turtle bycatch network are highlighted as directed red arrows here (arrows point to the one that was nominated). Node size and shading show the number of nominations each fisher received for sea turtle bycatch information (largest and darkest red = most nominations, small and grey = no nominations). Layout was set as a spring layout of edges across any network (to minimize overlap) and then expanded into a circular setting. See electronic supplementary material, figure S1 for illustrative comparisons of each fishing information type. (*c*) The observed in-degree assortativity (homophily) in comparison with the null distributions for the different information-sharing networks, and (*d*) the observed variance in the node eccentricity in comparison with the null distributions for the different information-sharing networks. Horizontal lines show the observed values from the actual networks (red = observed values are above the permutations, black = observed values are within the range of the permutations, purple = observed values are below the permutations). Polygon distributions show those generated by permutations (dark green = outgoing edge permutation that maintains the number of nominations each fisher makes, light green = edge permutation that maintains the number of nominations each fisher makes and also the number of times each individual was nominated). Due to differences in network factors, direct comparisons between the observed values are not informative. For details on each type of fishing information assessed refer to table 1.

bycatch with one another than expected, while this is not true for any other networks of information sharing. This conclusion also held when considering other measures of centrality. In electronic supplementary material, Information, we examined the variance in betweenness (as an alternative measure of centrality; electronic supplementary material, figure S3) and mean eccentricity for each network's fishers (rather than the variance; electronic supplementary material, figure S4). We also investigated the observed variance in node eccentricity in comparison with the null distributions (generated from the cross-network permutations; electronic supplementary material, figure S5) and the observed mean node eccentricity in comparison with the null distributions (electronic supplementary material, figure S6). The findings demonstrated that the sea turtle bycatch information-sharing network generally held some structural dissimilarities to all other fishing information-sharing networks assessed.

### 3.1.3. Cross-network correlations of dyadic ties

Gillnet skippers in our survey were asked to nominate individuals that they exchange useful information with about fishing and that they considered valuable to their fishing success. Respondents were then asked which topic of fishing information they talk to each nominated individual about (table 1). Given this system, we hypothesized that information-sharing networks across the assessed subtopics of fishing information would be correlated with one another, assuming that pairs of skippers (dyads) who share information within a specific network would be more likely to share information in

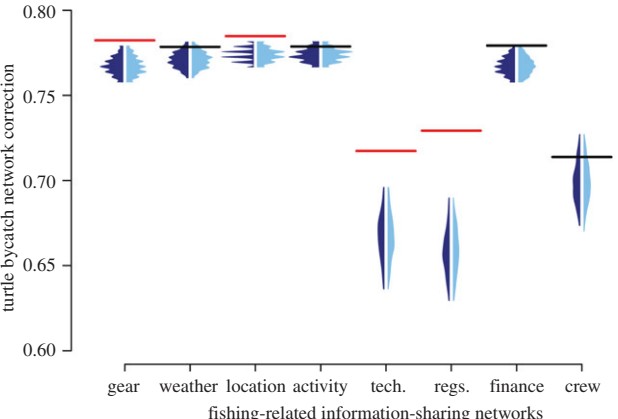

**Figure 3.** The observed correlation (and the correlations expected under the null models) between the sea turtle bycatch information-sharing network with all the other information networks. Horizontal lines show the observed values from the actual networks (red = observed values are above the permutations, black = observed values are within the range of the permutations). Polygon distributions show those generated by permutations (dark blue = network swap that maintains the number of nominations each individual makes and also the number of times each individual was nominated, but swaps the network these were made within while maintaining the number of times each network was nominated as overall, light blue = conservative network swap that is the same as dark blue, but also maintains the number of networks each dyad nominated each other for—but changes those networks (same as a gbi permutation but on the dyad-by-network edges). Comparison between networks can be made by comparing the distance between the observed values from the actual networks (horizontal lines) and their associated permutation distribution (polygon) to the distance between the observed and associated permutation for each network. Due to differences in network factors, direct comparisons between the observed values are not informative. For details on each type of fishing information assessed refer to table 1.

another network. As such, we expected all the other networks to significantly predict information-sharing within the network of particular interest (sea turtle bycatch information). Indeed, the sea turtle bycatch information-sharing network significantly correlated with all other networks (unfolded corr, $r = > 0.7$; standard $p < 0.01$). We also tested this observed correlation against that expected under the general social structure (cross-network null model 1—who gains information from whom overall; figure 1$c$) as well as controlling for the probability of nomination within each network (cross-network null model 2; figure 1$d$; see electronic supplementary material, Information). Under these null models, we found that the dyadic directed links within the sea turtle bycatch information-sharing network were significantly more correlated with four information-sharing networks (regarding gear, locations, technology and regulations—table 1) than expected under the general social structure (figure 3). Although the sea turtle bycatch information-sharing network held the highest raw correlation with networks of information regarding fishing locations (unfolded corr, $r = 0.78$), the largest difference between the correlation expected under the null models and the observed correlation was with information sharing regarding fishing regulations (unfolded corr, $r = 0.78$; mean expected corr cross-network null model 1, $r = 0.65$; mean expected corr cross-network null model 2, $r = 0.65$), suggesting that the fishing regulations network was particularly predictive of sea turtle bycatch information links given the underlying social structure of the system.

## 4. Discussion

By combining a fine-scale survey of a small-scale fishing community with a network null model approach that incorporates a pre-network data permutation procedure, we show that information-sharing networks about an issue of conservation concern (sea turtle bycatch) are dissimilar in degree assortativity and node eccentricity from other closely related information-sharing networks that relate to fishing (figure 2), more so than expected by simple differences in an individual's degree (how many people they are connected to). We also demonstrate that specific fishing information-sharing networks can still be predictive of how information about sea turtle bycatch is shared between fishers, even more so than expected under the nomination structure of who nominated whom (figure 3).

## 4.1. Structural differences between information-sharing networks

We found that the sea turtle bycatch network did not show any degree of assortativity (i.e. degree homophily—gillnet skippers talking to other gillnet skippers with a similar number of connections) despite the positive degree assortativity patterns across all other fishing information-sharing networks (figure 2c; electronic supplementary material, table S1). This finding indicates that certain mechanisms that drive information sharing between gillnet skippers in the other fishing information-sharing types assessed (and potentially social networks generally) may not be at play in the sea turtle bycatch information-sharing network [49,50]. The lack of discussion about sea turtle bycatch between gillnet skippers potentially indicates that sea turtle bycatches are not seen as something that warrants regular discussion in the San Jose gillnet skipper community. Indeed, previous research and field observations from the study site have suggested that fishers with higher bycatch rates tend not to put much effort into actively avoiding sea turtles captures unless they are specifically incentivized to do so (i.e. through the local not-for-profit's trial bycatch reduction initiative) [42]. Moreover, the possibility remains that fishers may take sea turtle bycatch and not discuss it with other fishers at all. Yet it may be precisely these types of fishers whose behaviour would be the ideal target for a sea turtle bycatch reduction intervention. Six out of 165 fishers surveyed in our study did not discuss sea turtle bycatch with any other fishers (figure 2b); however, all these fishers reported never catching sea turtles through direct questioning. To improve how information about the sea turtle bycatch reduction intervention is shared between fishers, the interventions managers could incorporate an educational discussion with fishers on the conservation status of sea turtle species captured in the fishery and provide information on the local variations in sea turtle bycatch rates prior to undertaking their planned expansion of the bycatch reduction strategy on trial. Additionally, other mechanisms could be expected to drive information sharing between fishers, prompting further research to investigate whether additional demographic characteristics correspond with different types of fishing information exchange. For example, several studies of small-scale fisheries have shown that similarity in gear type coincides with information exchange among fishers [30,80]. Similarly, longline fishers in Hawaii show a strong homophilic tendency for information exchange along ethnic lines [33].

We also found that the sea turtle bycatch information-sharing network has less variance in node centrality than expected, i.e. a more uniform individual-level network structure (figure 2d and table 2). The low variance in node eccentricity indicates that the sea turtle bycatch network has a more homogeneous network structure than the other networks (and many observed social networks, where high variability in node centrality is common and can result in highly nominated fishers forming [81,82]). This finding indicates that information about sea turtle bycatch will have less variation in the rate of diffusion throughout the San Jose skipper community, regardless of which skipper first started talking to other skippers in the community about the capture, compared with information-sharing in a network with higher variance in node eccentricity (e.g. the weather, fishing locations, fishing activity and finance).

As an addition to the above points, we found less variance in node centrality (figure 2d) and less variance in mean eccentricity (electronic supplementary material, figure S4) in the sea turtle bycatch information-sharing network when comparing with the cross-network null models (electronic supplementary material, figures S5 and S6). This lower variance shows that the variance and mean eccentricity is lower than expected, not just in comparison with the edge null models, but also lower than expected given the underlying social structure of who is connected to whom. This lower variance found when comparing the cross-network null models reinforces the hypothesis that the network's fine-scale structure (beyond who talks to whom) is contributing to these patterns. For example, certain personality traits that gillnet skippers hold, such as whether they would be willing to work with a local not-for-profit organization to implement sea turtle bycatch reduction strategies on their boats in future, may be contributing to skipper centrality within the network. This finding demonstrates a particularly interesting use of comparing results across various null models that randomize different processes.

The underlying assumption that a sea turtle bycatch information-sharing network might be a better target for transmitting information about the sea turtle bycatch reduction intervention's existence and aims over other relevant topics of fishing information (e.g. fishing location, vessel technology and maintenance) or a more general 'fishing' information-sharing network warrants further investigation. A central consideration is the desired goal underpinning the transmission of information about the sea turtle bycatch reduction intervention's existence and aims. There is a rapidly growing body of evidence that suggests information frequently spreads as 'simple contagions' and behaviours spread

as 'complex contagions' [25,83–86]. The complexity of the contagion holds significant ramifications for the modes and extent of transmission [87]. If resource managers working in this study's fishing system would like to understand the mode and extent of information spread about the sea turtle bycatch reduction intervention's existence and aims across the sea turtle bycatch information-sharing network, then simulating simple contagion where transmission occurs between individuals that are socially connected to one another could inform them of the expected rate and extent of transmission that messages relevant to their intervention might have across this network. Simple contagion modelling could also be compared across other specific fishing information sharing types that might be associated with the intervention (e.g. fishing finances, vessel technology and maintenance) and the more general 'fishing' network to better understand how specific information types, or combinations of, affect the mode and extent of transmission of messages relevant to the intervention in question. However, if resource managers were interested in understanding the mode and extent of adoption of the sea turtle bycatch intervention in the community, simulating complex contagion that involves some 'complexity' beyond the raw number of social ties to informed individuals would be a more informative strategy [25]. For example, for fishers to change their fishing practices, they may require social reinforcement via multiple-trusted contacts [25,88].

## 4.2. Cross-network correlations of dyadic links

Understanding correlations between networks allows for assessing fisher-to-fisher (dyadic link) information-sharing differences between multiple networks. The similarity identified between the fine-scale structures of the information-sharing networks assessed demonstrates that relying on simple network measures without the use of the null model comparisons could potentially result in an improper assessment of network structure. Moreover, insight into these differences helps identify social contexts suited to conservation interventions and, more broadly, offers insight into the generalizability of network research [89].

We demonstrate that across all the networks assessed, the fine-scale structures of the fisher's information-sharing networks are more similar than otherwise expected based on the number of links or even who is linked to whom. While this similarity assures that in the current study's gillnet skipper network, knowledge about a social network based on general information spread should be transferable into understanding how novel information spreads. We also show the networks that are most closely related to the specific network of conservation interest, offering a greater understanding of how information flows relevant to the broader topic of information-sharing about fishing are structured and relate to one another (figure 3).

Our results indicate that the fishing regulations network, followed by the vessel technology and maintenance, gillnet type and maintenance, and fishing location networks, are more correlated with the sea turtle bycatch network structure than expected under the cross-network null models (figure 3). This finding gives insight into how fishers perceive information relating to sea turtle bycatch. For example, the correlation between sea turtle bycatch and the fishing regulation network could be because fishers perceive sea turtle bycatch as something they must abide by, similar to fishing regulations (related to the business and governance of fishing; table 1). This correlation is supported by a supplementary structural analysis that shows that the sea turtle bycatch and regulation networks are structurally dissimilar concerning node variance to all other information sharing (electronic supplementary material, figures S3, S9 and S10). Moreover, the correlations identified between sea turtle bycatch and the topics of vessel technology and maintenance, fishing gear and fishing location indicate a perception of sea turtle bycatch as part of the process of fishing (table 1). While these results begin to provide a more in-depth insight into how sea turtle bycatch information-sharing relates to other type of fishing information and how this information is perceived by fishers, further exploration is needed to determine the process underlying the structural differences identified.

# 5. Conclusion

We quantified the underlying structure of a small-scale fishery social system across nine information-sharing networks relating to fishing. Our study demonstrates how networks of information-sharing regarding a conservation-relevant topic (sea turtle bycatch) are structurally dissimilar in degree assortativity and node eccentricity from other types of fishing information-sharing, and the extent to which fisher–fisher (dyadic) ties can be correlated with other information-sharing networks. The lack

of degree assortativity identified among fishers sharing sea turtle bycatch information may suggest that a rapid diffusion of information about the planned intervention could be less likely as highly nominated fishers may often not discuss sea turtle bycatch with other highly nominated fishers. The low variance in node centrality identified within the same network may suggest that resource managers for instance could place less emphasis on which fishers they choose to start seeding information with about the intervention, as individuals have similar connectivity anyway. Finally, resource managers could also consider using the data comparing fishing information types to gain insight into this fisher's perception of sea turtle bycatch to inform engagement processes as part of the implementation of behaviour-change interventions. Our results also show how social network approaches can be useful for identification of the extent of structural differences between networks and provide information about which other networks are best correlated with the conservation-relevant information sharing. Together these findings contribute understanding to how fine-scale differences in information shared between resource users can influence network structure and what implications this might have for conservation interventions.

Ethics. Documented, free, prior and informed consent was sought from all respondents before they could take part in the study. This research has Research Ethics Approval (CUREC 1A; Ref No: R52516/RE001 and R52516/RE002).

Data accessibility. The data and R scripts that support the findings of this study are available at https://github.com/JoshFirth/bycatch_information_flow.

Authors' contributions. W.N.S.A. and E.J.M.-G. designed the study, and W.N.S.A and J.A.F. wrote the first draft. W.N.S.A., E.J.M.-G., B.I.-E. J.A.-S. and J.C.M. contributed to survey design. W.N.S.A. and B.I.-E. collected the data. J.A.F. and W.N.S.A. carried out the analysis. W.N.S.A., E.J.M.-G., and J.A.F. interpreted the data and planned the draft. All authors contributed significantly to revising the manuscript.

Competing interests. We have no competing interests.

Funding. W.N.S.A. was supported by a PhD Scholarship from the Commonwealth Scholarship Commission in the UK and the University of Oxford (PhD scholarship NZCR-2015-174), the OX/BER Research Partnership Seed Funding Fund in collaboration with the University of Oxford and Humboldt-Universität zu Berlin (OXBER_STEM7), and acknowledges funding from the Pew Charitable Trusts through a Pew Fellowship to E.J.M.-G. J.A.F was supported by a research fellowship from Merton College and BBSRC (grant no. BB/S009752/1) and acknowledges funding from NERC (grant no. NE/S010335/1).

Acknowledgements. We thank the US National Ocean and Atmospheric Administration, National Marine Fisheries Service, Southwest Fisheries Science Center for supporting this research. A special thanks to David Sarmiento Barturen and Natalie Bravo for your support during data collection. We thank Robert Arlinghaus and the anonymous reviewers whose comments greatly improved the manuscript.

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
