## [Peer Review File · Royal Society Open Science]

Review History

RSOS-211240.R0 (Original submission)

Review form: Reviewer 1

Is the manuscript scientifically sound in its present form?

Yes

Are the interpretations and conclusions justified by the results?

Yes

Is the language acceptable?

Yes

Do you have any ethical concerns with this paper?

No

Have you any concerns about statistical analyses in this paper?

Yes

Recommendation?

Major revision is needed (please make suggestions in comments)

Comments to the Author(s)

The author have replied to my previous queries. I still have serious doubts on assessing similarity of networks via the Mantell Test, I think that the use of graphlets as described in Przuyl 2007 and Tantardini et al. 2019 would be a better approach. Except this, I also think that the permutation exercise is interesting but lacks the nuances of more "constraining" null models that maintain, for example, the number of triangles in the network. Still it can work as a coarse grained assessment of whether certain network metrics are more or less expected than by chance.

I have no other comments, but two minor points that relate to language:

Both on page 21 (out of 36)

Lines: 47- 48: i think the authors need to be more clear. In fact, randomness should be represented by an assortativity coefficient of 0. Disassortative networks are the ones in which high degree nodes tend to connect to low degree nodes... and assortative networks the ones in which high degree nodes tend to connect to other high degree nodes.

Line 60: something is missing here... i think it should read: peripheral fishers would be more likely to be connected to other peripheral fishers.

Review form: Reviewer 2**Is the manuscript scientifically sound in its present form?**

Yes

Are the interpretations and conclusions justified by the results?

No

Is the language acceptable?

Yes

Do you have any ethical concerns with this paper?

No

Have you any concerns about statistical analyses in this paper?

No

Recommendation?

Major revision is needed (please make suggestions in comments)

Comments to the Author(s)

This manuscript assesses the information-sharing network in a small-scale fishery. The authors build different information-sharing networks based on the type of information that is shared by fishers (turtle bycatch, where to fish, regulation, etc.), and compare their structure using permutation-based null models and two key metrics: degree assortativity and node eccentricity. The authors use their results to highlight how differences in the structure of how different types of information that is shared between fishers, have implications for the diffusion of conservation interventions.

The article is well written and succinct and the topic would be of interest to this journal's audience. There are two main concerns I have with this paper. The first concern is that it is not

clear why the measures selected (degree assortativity and node eccentricity) are appropriate to address the research question in the particular context. They cite a couple of studies of small scale fisheries to support their claim that degree assortativity is “known to have important social implications for the operation and emergence of competition and cooperation”, but these two studies actually assess homophily based on actor attribute (e.g. gear type used), which is very different to the type of homophily effect captured by degree assortativity (which is based on the centrality of the actor). Other than this, in Table 2 they authors cite ecological studies using the measure. In sum, I would expect a much better empirical, theoretical, support for their choice of metric.

The second major concern I have is that I am not sure that the authors’ conclusions can be justified by their results. Specifically, the authors conclude that information-sharing networks about turtle bycatch are “structurally dissimilar” from the other information-sharing networks and, even more troubling – they conclude that the “usual mechanisms that drive information sharing between fishers in the other fishing-information types assessed (and potentially social networks generally) are not at play in the turtle bycatch information-sharing network”. The problem I have with this conclusion is that degree assortativity is only one of many mechanisms that could be driving the formation of these information-sharing networks, and unfortunately, with the approach employed, there is no way that the authors can ascertain what are the mechanisms driving network formation (more on this below). My initial reaction was that in order to address this, the authors could refocus their conclusions to talk specifically about the observed differences in degree assortativity (rather than structure in general), but even then, the problem is that some, even more basic mechanisms of network formation (that are not accounted for in the study) could actually explain the differences observed in degree assortativity. For example, reciprocity is commonly observed in information-sharing networks, and from what I can see in Figure 4 there seems to be a fair amount of reciprocity in the data. However, the authors do not mention these and other potential mechanisms, how they have accounted for them or why they are not relevant to their analysis or context (for example, homophily based on demographic characteristics such as race or family ties/clans, have been found as key mechanisms of network formation in fishers information-sharing networks). In other words, I think the authors have to do a much better job of supporting their methodological approach. If they cannot fully support their choice (and right now I don’t think the support required is there), I would invite the authors to re-run their results using appropriate methods for the structural analysis of networks where multiple competing mechanisms can be tested concurrently.

Related to my last point above, I note that a previous reviewer highlighted ERGMs as a more stringent null model that is able to take into account multiple network-formation effects. In my opinion, the authors’ response to this major concern (as highlighted by the reviewer) is not convincing. The authors argue that while they acknowledge that ERGMs can perform many of the functions that their permutation null model approach can, they think there is still novelty in applying the permutation-based null model because they are only aware of 2 studies on human networks that use the same permutation null model that they use. This is not convincing for two reasons. First, there are tons of permutation-based models that have been used in the analysis of human networks since at least the 80s. They all vary slightly in what the null models are conditioned on, but they are all doing the same thing i.e. trying to get a test statistic for the measure of interest by comparing it to a baseline. So saying that “there is novelty in applying these analytical methods” is not enough. Second, the point about ERGMs is not that they can perform many of the functions that permutations test can. The point is that these models can perform all of the functions that the permutation model applied by the authors can, plus many more. ERGMs are much more advanced models that seem to me are much more suitable for the task at hand. This not to say that permutation null models are not adequate for certain tasks. But given that the authors have not fully supported why it makes sense to ONLY compare the structure of the networks on the basis of their degree assortativity and, also importantly, why the particular permutations implemented are suitable given the particular context and type of data, I am afraid I am not convinced that the permutation null models selected are suitable.

I provide minor comments in the attached pdf document (Appendix A).

Decision letter (RSOS-211240.R0)

Dear Dr Arlidge

The Editors assigned to your paper RSOS-211240 "Assessing information-sharing networks within small-scale fisheries and the implications for conservation interventions" have now received comments from reviewers and would like you to revise the paper in accordance with the reviewer comments and any comments from the Editors. Please note this decision does not guarantee eventual acceptance.

Please submit your revised manuscript and required files (see below) no later than 21 days from today's (ie 03-Sep-2021) date. Note: the ScholarOne system will 'lock' if submission of the revision is attempted 21 or more days after the deadline. If you do not think you will be able to meet this deadline please contact the editorial office immediately.

on behalf of Prof Pete Smith (Subject Editor)
openscience@royalsociety.org

Associate Editor Comments to Author:

The two reviewers have offered a range of comments that you should address in this revision. While one of the reviewers expresses concerns regarding the utility/value of the approaches adopted, if you can persuade the editors and reviewers that your paper has archival value here (rather than perhaps being a paradigm-shifting approach), the journal would be able to accept it for publication: we do not require ground-breaking novelty, but there should be some purpose to the work if it is to be published as archivally useful. Good luck with the revisions, and we'll look forward to receiving these in due course. All best.

Reviewer comments to Author:

Reviewer: 1

Comments to the Author(s)

The author have replied to my previous queries. I still have serious doubts on assessing similarity of networks via the Mantell Test, I think that the use of graphlets as described in Przuyl 2007 and Tantardini et al. 2019 would be a better approach. Except this, I also think that the permutation exercise is interesting but lacks the nuances of more "constraining" null models that maintain, for example, the number of triangles in the network. Still it can work as a coarse grained assessment of whether certain network metrics are more or less expected than by chance.

I have no other comments, but two minor points that relate to language:

Both on page 21 (out of 36)

Lines: 47- 48: i think the authors need to be more clear. In fact, randomness should be represented by an assortativity coefficient of 0. Disassortative networks are the ones in which high degree nodes tend to connect to low degree nodes... and assortative networks the ones in which high degree nodes tend to connect to other high degree nodes.

Line 60: something is missing here... i think it should read: peripheral fishers would be more likely to be connected to other peripheral fishers.

Reviewer: 2

Comments to the Author(s)

This manuscript assesses the information-sharing network in a small-scale fishery. The authors build different information-sharing networks based on the type of information that is shared by fishers (turtle bycatch, where to fish, regulation, etc.), and compare their structure using permutation-based null models and two key metrics: degree assortativity and node eccentricity. The authors use their results to highlight how differences in the structure of how different types of information that is shared between fishers, have implications for the diffusion of conservation interventions.

The article is well written and succinct and the topic would be of interest to this journal's audience. There are two main concerns I have with this paper. The first concern is that it is not clear why the measures selected (degree assortativity and node eccentricity) are appropriate to address the research question in the particular context. They cite a couple of studies of small scale fisheries to support their claim that degree assortativity is "known to have important social implications for the operation and emergence of competition and cooperation", but these two studies actually assess homophily based on actor attribute (e.g. gear type used), which is very different to the type of homophily effect captured by degree assortativity (which is based on the centrality of the actor). Other than this, in Table 2 they authors cite ecological studies using the measure. In sum, I would expect a much better empirical, theoretical, support for their choice of metric.

The second major concern I have is that I am not sure that the authors' conclusions can be justified by their results. Specifically, the authors conclude that information-sharing networks about turtle bycatch are "structurally dissimilar" from the other information-sharing networks

and, even more troubling – they conclude that the “usual mechanisms that drive information sharing between fishers in the other fishing-information types assessed (and potentially social networks generally) are not at play in the turtle bycatch information-sharing network”. The problem I have with this conclusion is that degree assortativity is only one of many mechanisms that could be driving the formation of these information-sharing networks, and unfortunately, with the approach employed, there is no way that the authors can ascertain what are the mechanisms driving network formation (more on this below). My initial reaction was that in order to address this, the authors could refocus their conclusions to talk specifically about the observed differences in degree assortativity (rather than structure in general), but even then, the problem is that some, even more basic mechanisms of network formation (that are not accounted for in the study) could actually explain the differences observed in degree assortativity. For example, reciprocity is commonly observed in information-sharing networks, and from what I can see in Figure 4 there seems to be a fair amount of reciprocity in the data. However, the authors do not mention these and other potential mechanisms, how they have accounted for them or why they are not relevant to their analysis or context (for example, homophily based on demographic characteristics such as race or family ties/clans, have been found as key mechanisms of network formation in fishers information-sharing networks). In other words, I think the authors have to do a much better job of supporting their methodological approach. If they cannot fully support their choice (and right now I don't think the support required is there), I would invite the authors to re-run their results using appropriate methods for the structural analysis of networks where multiple competing mechanisms can be tested concurrently.

Related to my last point above, I note that a previous reviewer highlighted ERGMs as a more stringent null model that is able to take into account multiple network-formation effects. In my opinion, the authors' response to this major concern (as highlighted by the reviewer) is not convincing. The authors argue that while they acknowledge that ERGMs can perform many of the functions that their permutation null model approach can, they think there is still novelty in applying the permutation-based null model because they are only aware of 2 studies on human networks that use the same permutation null model that they use. This is not convincing for two reasons. First, there are tons of permutation-based models that have been used in the analysis of human networks since at least the 80s. They all vary slightly in what the null models are conditioned on, but they are all doing the same thing i.e. trying to get a test statistic for the measure of interest by comparing it to a baseline. So saying that “there is novelty in applying these analytical methods” is not enough. Second, the point about ERGMs is not that they can perform many of the functions that permutations test can. The point is that these models can perform all of the functions that the permutation model applied by the authors can, plus many more. ERGMs are much more advanced models that seem to me are much more suitable for the task at hand. This not to say that permutation null models are not adequate for certain tasks. But given that the authors have not fully supported why it makes sense to ONLY compare the structure of the networks on the basis of their degree assortativity and, also importantly, why the particular permutations implemented are suitable given the particular context and type of data, I am afraid I am not convinced that the permutation null models selected are suitable.

I provide minor comments in the attached pdf document.

===PREPARING YOUR MANUSCRIPT===

===PREPARING YOUR REVISION IN SCHOLARONE===

- If you are providing image files for potential cover images, please upload these at this step, and inform the editorial office you have done so. You must hold the copyright to any image provided.
- A copy of your point-by-point response to referees and Editors. This will expedite the preparation of your proof.

- Ensure that your data access statement meets the requirements at <https://royalsociety.org/journals/authors/author-guidelines/#data>. You should ensure that you cite the dataset in your reference list. If you have deposited data etc in the Dryad repository, please include both the 'For publication' link and 'For review' link at this stage.
- If you are requesting an article processing charge waiver, you must select the relevant waiver option (if requesting a discretionary waiver, the form should have been uploaded at Step 3 'File upload' above).
- If you have uploaded ESM files, please ensure you follow the guidance at <https://royalsociety.org/journals/authors/author-guidelines/#supplementary-material> to include a suitable title and informative caption. An example of appropriate titling and captioning may be found at https://figshare.com/articles/Table_S2_from_Is_there_a_trade-off_between_peak_performance_and_performance_breadth_across_temperatures_for_aerobic_scope_in_teleost_fishes_/3843624.

Author's Response to Decision Letter for (RSOS-211240.R0)

See Appendix B.

RSOS-211240.R1 (Revision)

Review form: Reviewer 1

Is the manuscript scientifically sound in its present form?

Yes

Are the interpretations and conclusions justified by the results?

Yes

Is the language acceptable?

Yes

Do you have any ethical concerns with this paper?

No

Have you any concerns about statistical analyses in this paper?

No

Recommendation?

Accept as is

Comments to the Author(s)

I would like to thank the author for engaging with all my previous comments. I do think that all queries were addressed.

Decision letter (RSOS-211240.R1)

Dear Dr Arlidge,

It is a pleasure to accept your manuscript entitled "Assessing information-sharing networks within small-scale fisheries and the implications for conservation interventions" in its current form for publication in Royal Society Open Science. The comments of the reviewer(s) who reviewed your manuscript are included at the foot of this letter.

on behalf of Pete Smith (Subject Editor)
openscience@royalsociety.org

Associate Editor Comments to Author:
Comments to the Author:
Congratulations on the success of your paper!

Reviewer comments to Author:
Reviewer: 1

Comments to the Author(s)

I would like to thank the author for engaging with all my previous comments. I do think that all queries were addressed.

Appendix A**ROYAL SOCIETY
OPEN SCIENCE****Assessing information-sharing networks within small-scale fisheries and the implications for conservation interventions**

Journal:	Royal Society Open Science
Manuscript ID	RSOS-211240
Article Type:	Research
Date Submitted by the Author:	29-Jul-2021
Complete List of Authors:	Arlidge, William; Leibniz-Institute of Freshwater Ecology and Inland Fisheries in the Forschungsverbund Berlin eV, Biology and Ecology of Fishes; Firth, Josh; University of Oxford, Department of Zoology Alfaro-Shigueto, Joanna; Pro Delphinus; University of Exeter, School of Biosciences Ibanez-Erquiaga, Bruno; Universidad San Ignacio de Loyola, Departamento de Química y Biología; Asociación CONSERVACION Mangel, Jeffrey; ProDelphinus; University of Exeter, School of Biosciences Squires, Dale; Southwest Fisheries Science Center, Economics and Social Science Division Milner-Gulland, EJ; University of Oxford Mathematical Physical and Life Sciences Division, Zoology
Subject:	environmental science < BIOLOGY, behaviour < BIOLOGY
Keywords:	Bycatch, null model, permutation, social network analysis, sea turtle
Subject Category:	Ecology, Conservation, and Global Change Biology

Author-supplied statements

Relevant information will appear here if provided.

Ethics

Does your article include research that required ethical approval or permits?:

Yes

Statement (if applicable):

Documented, free, prior, and informed consent was sought from all respondents before they could take part in the study. This research has Research Ethics Approval (CUREC 1A; Ref No: R52516/RE001 and R52516/RE002).

Data

It is a condition of publication that data, code and materials supporting your paper are made publicly available. Does your paper present new data?:

Yes

Statement (if applicable):

The data and R scripts that support the findings of this study are available at https://github.com/JoshFirth/bycatch_information_flow.

Conflict of interest

I/We declare we have no competing interests

Statement (if applicable):

CUST_STATE_CONFLICT :No data available.

Authors' contributions

This paper has multiple authors and our individual contributions were as below

Statement (if applicable):

W.N.S.A. and E.J.M-G. designed the study, and W.N.S.A, and J.A.F. wrote the first draft. W.N.S.A., E.J.M-G., B.I.E., J.A.S., and J.C.M. contributed to survey design. W.N.S.A. and B.I.E. collected the data. J.A.F. and W.N.S.A. carried out the analysis. W.N.S.A., E.J.M-G., and J.A.F. interpreted the data and planned the draft. All authors contributed significantly to revising the manuscript.

Arlidge et al.

29 July 2021

Royal Society Open Science – Response to Proceedings of the Royal Society B

We have addressed the two expert reviewer's comments from our submission to the Proceedings of the Royal Society B in detail. For each point raised, the reviewer's comments are provided followed by our response in **bold**.

Reviewers' comments:

Reviewer #1 (Remarks to the Author):

Comments to the Author(s)

Sociology terms, such as nodes and network, may be replaced with words in the context of this paper. Using the jargon somewhat distracts the real issue. For example, "nodes" can be replaced with "fishers" and "network" with "information type." I point out this and other suggestions in the following.

Response: We thank the Reviewer for their helpful suggestions that have helped to improve the manuscript. We have made an effort to reduce the network science jargon through the manuscript. The term "nodes" has been replaced with "fishers" when discussing respondents in the survey, and the term "network" has been replaced with "information type" in multiple areas of text. However, we have retained "node" and "network" in some instances that we felt were applicable such as when referring to a known network metric like node eccentricity. We have ensured that the terms "node" and "network" are clearly defined in the text.

Major points:

Abstract

This abstract is a little "weak." The second sentence poses the existing question/problem: "However, it remains unclear how fine-scale differences in information shared between resource users can influence network structure and the success of behaviour-change interventions." And the concluding statement at the end of the abstract states that ... "This finding highlights that fine-scale differences in the information shared between resource users may influence network structure." Just reading the abstract, it appears that no new information was gained from this study - it was known that fine-scale differences in the information shared between resource users may influence network structure (second sentence) but it was unknown how the differences would affect network structure. I suggest editing the abstract and adding more results to make it more convincing.

Response: The Reviewer highlights several good points concerning the clarity of the abstract. We have included the following sentence presenting additional results from the study:

"We also demonstrate that patterns of information sharing between fishers related to sea turtle bycatch are more similar to information sharing about

fishing regulations, and vessel technology and maintenance, than to weather, fishing activity, finances, and crew management.”

The abstract is written in a rather abstract form where there is no specific conclusion with respect to turtle bycatch and information sharing among fishers is stated. Only statement is “... the general network structure detailing information sharing about sea turtle bycatch differs from other fishing-related information sharing, specifically in degree assortativity (homophily) and eccentricity.” I think it would be useful to provide a sentence or two about how it is different from other information sharing.

Response: We have included specifics on how the sea turtle bycatch information-sharing network differs from the other information-sharing networks assessed. The relevant sentence reads:

“Specifically, no significant degree assortativity (homophily) was identified and the variance in node eccentricity was lower than expected under our null models”

Also, it may be good to add a sentence about how that finding can be used in turtle conservation.

Response: We have included a sentence about how that finding can be used in sea turtle conservation:

“Our findings highlight the importance of assessing information-sharing networks in contexts directly relevant to the desired intervention and demonstrate the identification of social contexts which might be more or less appropriate for information-sharing related to planned conservation actions.”

Line 37: “The conservation and management of common-pool natural resources such as fisheries often involve behaviour-change interventions with resource users.”

This first sentence seems to be convoluted. Perhaps, my definition of “common-pool resources” may be different from the authors’. I think “common-pool natural resources” are resources that are shared by many, e.g., fishes. With this definition, fisheries are not common-pool natural resources themselves but they harvest common-pool resources. If there is a different definition for “common-pool natural resources,” it should be clearly stated. Furthermore, fisheries do not require conservation. I think the sentence may be split into two sentences, where one is about conservation of natural resources and the other about management of fisheries that harvest natural resources.

Because of the issue I mentioned in the previous paragraph, the first paragraph is a little difficult to follow. I think the authors are trying to state that behavior-changing interventions are sometimes necessary to successfully manage small-scale fisheries. I suggest rewriting this paragraph. Perhaps, focus on fishery management in this paragraph without considering the bycatch issues.

Alternatively, start the manuscript from the second paragraph and eliminate the first paragraph.

Response: We have clarified the text around common-pool natural resources and the harvesters of these resources. We have also refined the first paragraph to only talk about fisheries management and talk about the bycatch issues in the following paragraph. The first paragraph now reads:

“Managing the harvesting of common-pool natural resources such as fishes often involves behaviour-change interventions with resource harvesters [1, 2]. These can include interventions like the enforcement of rules, social marketing, and education campaigns [3-5]. Informed by other behavioural-change disciplines like public health and social marketing, conservation science is increasingly looking to understand the social structure of fishing communities targeted for interventions to predict how information flows influence the transmission of pro-environmental behaviours [6, 7]. However, understanding how fine-scale differences in information shared between resource users can influence network structure and what implications this might have for conservation interventions has received little research attention.”

Line 67: “... in enacted” should be “... is enacted.”

Response: Done.

Line 72: “We apply null models that incorporate pre-network data permutations to fisher information-sharing data, to explore a potentially crucial, but currently untested assumption when analysing social networks in conservation science and natural resource management”

Maybe it is better to state “we apply the permutation-based null model approach to fisher information-sharing data in order to explore...,” or something similar?

Response: This paragraph has been rearranged. The relevant sentence has been moved to the final paragraph of the introduction and reads:

“In this study, we apply a social network analysis and permutation-based null model approach to assess whether networks of small-scale fisher’s information sharing about sea turtle bycatch are structurally similar to other types of fishing information-sharing networks (Table 1).”

Materials and methods

Line 161: “In each network, the nodes were the individuals, and the binary directed edges were the nominations by one node (sender) of another node (receiver) for this information type.”

Is it possible to keep “individuals” (or “fishers” or “skippers”) rather than changing the word to “nodes” throughout the manuscript? Calling individual fishers “nodes” makes the manuscript seem to be detached from the real conservation issues.

**Response: The majority of references to “node(s)” has been changed to “fisher(s)” or**
**“individual(s)” throughout the manuscript (depending on whether the text is talking**
**specifically about the surveyed fishers, or individual people more generally). The exception**
**for inclusion of “node” is when referring to a known metric that uses this term i.e., “node**
**centrality” and “node eccentricity”.**

Line 174: “While the null model methods applied in the current study were developed in
ecology, they are beginning to be used in human network analysis. For example, in the fields of
epidemiology for assessing human contact tracing disease control measures [39].”

These sentences should be moved to Discussion.

**Response: We have moved the sentence in question to the second paragraph of the**
**‘Structural differences across information-sharing networks’ section in the methods, which**
**was formerly was a paragraph in the supplementary materials. The paragraph reads:**

**“Network null models (routines that generate different types of null datasets**
**against which the observed dataset can be compared) are a group of statistical**
**models commonly applied in network analysis. Specifically, null models are**
**especially useful when investigating hypotheses in datasets, control groups are**
**difficult to establish, exogenous treatments are unavailable, and observations**
**may be missing or biased [48-50]. As such, null model methods are important**
**because network data is comprised of non-independent observations of multiple**
**individuals, and small variations in how data are collected between respondents**
**can easily generate patterns that appear as social structure [50, 51]. Null models**
**have been applied to network data in sociology since the 1970s [48] and**
**discipline-specific developments have subsequently been made to statistical**
**models such as exponential random graph models [52, 53], conditional uniform**
**graph tests [54-56] and quadratic assignment procedure tests [57-59]. Since the**
**mid-1990s, the field of ecology has also made extensive use of null models to**
**develop specialised hypothesis testing routines and treat underlying uncertainty**
**or data collection methodology biases when interrogating non-human animal**
**network data [60-62]. Here we expand the application of the permutation-based**
**null model approach routinely used in ecology to human social networks, which**
**have also been applied in the field of epidemiology for assessing human contact**
**tracing disease control measures [63].”**

Line 181: “The degree assortativity (or homophily) coefficient measures the extent to which
central nodes are connected to other central nodes, and peripheral nodes are connected to other
peripheral nodes based on a particular trait.”

When replacing “nodes” with either “individuals” or “fishers,” this sentence becomes more
understandable (at least to me): “The degree assortativity (or homophily) coefficient measures
the extent to which central fishers are connected to other central fishers, and peripheral fishers

Arlidge et al.

29 July 2021

Royal Society Open Science – Response to Proceedings of the Royal Society B

are connected to other peripheral fishers based on a particular trait, such as....”

Response: “Nodes” have been changed to “fishers” in this sentence as suggested.

Line 187: “When nodes of similar centrality are randomly distributed in a network (i.e., fully disassorted), those networks do not always score -1 due to the minimum value depending on the number of node types and the relative number of links within each group [40]”

Again, replacing “nodes” with “fishers” makes the sentence more understandable. One question: what are “node types” in this sentence? What “types” of fishers are you referring to?

Response: “Nodes” have been changed to “fishers” in this sentence, following the reviewer’s earlier suggestion. The sentence now reads:

“When fishers of similar centrality are randomly distributed in a community (i.e., disassorted), those networks do not always score -1 due to the minimum value depending on the number of fishers and the relative number of ties within each group [64].”

Line 231: “The first edge permutation simply allowed the randomisation of all in-going links, while maintaining the number of nominations (out-going links) each individual made within this information-sharing network”

In other places, the authors used “in-coming” rather than “in-going,” e.g., Table S1. I suggest using “in-coming” (or “in-going”) throughout the text.

Response: All references to “in-going” in the text have been changed to “incoming”.

Results

Means are provided in the results section. Were scores (networks, links, etc.) symmetric? If not, medians may be a better statistic to describe the central tendency.

Response: Network links were not symmetric across each information-sharing network assessed. We list the total number of links across each network in Table S1(b) of the supplementary materials.

Line 293: “As such, there was no evidence for a non- random tendency for highly nominated nodes to be disproportionately connected to other highly nominated nodes, nor for rarely nominated nodes to be disproportionately connected to other rarely nominated nodes.”

Here is another example in which using the word “nodes” makes a sentence less understandable. At least in my mind, when I replace “nodes” with “fishers”, the sentence is a little more digestible.

Response: “Nodes” have been changed to “fishers” in this sentence. The sentence now reads:

**“As such, there was no evidence for a non-random tendency for highly**
**nominated fishers to be disproportionately connected to other highly nominated**
**fishers, nor for rarely nominated fishers to be disproportionately connected to**
**other rarely nominated fishers.”**

9 **Discussion**

Line 413: “We demonstrate that across all the networks assessed, the fine-scale structures of our
information-sharing networks are more similar than otherwise expected based on the number of
links or even who is linked to whom.”

I got a bit confused about the term “our information-sharing networks.” Should this be “fisher’s
information-sharing network”? Or is the inference about a broader population?

**Response: The Reviewer is correct, the use of “fisher’s information-sharing network” is**
**more appropriate. We have implemented the suggested change. The sentence now reads:**

**“We demonstrate that across all the networks assessed, the fine-scale structures**
**of the fisher’s information-sharing networks are more similar than otherwise**
**expected based on the number of links or even who is linked to whom.”**

Line 417: “..., the similarity also demonstrates that relying on simple network measures without
the use of the null model comparisons could potentially results in an improper assessment of
network structure.”

Did the authors actually compare between conclusions with and without the null model
comparisons? I don’t recall seeing them.

**Response: The comparison between simple network measures without the use of null**
**models and null models is implicit in the analysis undertaken, we now hope we have made**
**this clear in the text. Metric measures are undertaken and presented as the horizontal line**
**and the expected distribution of the measure from the null models as coloured polygons,**
**demonstrating whether the observed measure is above, within, or below the expected null**
**distribution. So, each observed statistic is always compared to the null model, and this is**
**what determines whether it is higher, or lower (or not significantly different) from**
**expected.**

The paragraph mixes discussions about the findings of the study and methods (null model
comparisons). I suggest separating the discussion about the methods from this paragraph. That
discussion (null model comparisons) may be provided first in the discussion section.

**Response: We agree that separating the study findings from the methods makes the overall**
**discussion clearer. The first two paragraphs of the discussion now read:**

**“Understanding correlations between networks allows for assessing fisher-to-**
**fisher (dyadic link) information-sharing differences between multiple networks.**
**The similarity identified between the fine-scale structure of the information-**

**sharing networks assessed demonstrates that relying on simple network**
**measures without the use of the null model comparisons could potentially result**
**in an improper assessment of network structure. Moreover, insight into these**
**differences helps identify social contexts suited to conservation interventions,**
**and more broadly, offers insight into the generalisability of network research**
**[80].**

**We demonstrate that across all the networks assessed, the fine-scale structures**
**of the fisher’s information-sharing networks are more similar than otherwise**
**expected based on the number of links or even who is linked to whom. While**
**this similarity assures that in the current study’s gillnet skipper network,**
**knowledge about a social network based on general information spread should**
**be transferable into understanding how novel information spreads. We also**
**show the networks that are most closely related to the specific network of**
**conservation interest, offering a greater understanding of how information**
**flows relevant to the broader topic of information-sharing about fishing are**
**structured and relate to one another (Fig. 3).”**

Line 441: “Our study demonstrates how networks of information-sharing regarding a
conservation-relevant topic (sea turtle bycatch) are structurally dissimilar from other fishing-
related information-sharing networks, and the extent to which dyadic links can be non-randomly
predicted from other information-sharing networks.”

Given this finding, how would resource managers use information sharing structures among the
gillnet fishers to reduce turtle bycatch in this region?

**Response: We have included the following text discussing how the study’s findings above**
**can support resource managers reduce sea turtle bycatch in the region:**

**“The lack of degree assortativity (homophily) identified among fishers sharing**
**sea turtle bycatch information may suggest that a rapid diffusion of information**
**about the planned intervention could be less likely as highly nominated fishers**
**may often not discuss sea turtle bycatch with other highly nominated fishers. The**
**low variance in node centrality identified within the same network may suggest**
**that resource managers for instance could place less emphasis on which fishers**
**they choose to start seeding information with about the intervention, as**
**individuals have similar connectivity anyway. Finally, resource managers could**
**also consider using the data comparing fishing information types to gain insight**
**into these fisher’s perception of sea turtle bycatch to inform engagement**
**processes as part of the implementation of behaviour change interventions.”**

We also added a sentence in the preceding paragraph of the discussion that informs
the final sentence of the newly added text in the conclusion:

**“Moreover, the correlations identified between sea turtle bycatch and the**
**topics of vessel technology and maintenance, fishing gear, and fishing**

location indicate a perception of sea turtle bycatch as part of the process of fishing (Table 1).”

Figure 1. Why are there eight nodes when calculating test statistic but only five nodes in other steps? I’m thoroughly confused looking at these plots - A/B and C/D are identical in the structure as represented in the figure. Are they supposed to be the same?

Response: Eight nodes were used to portray the calculation of the test statistic as a five node network was thought to be more difficult to visually represent assortativity. However, we see how varying the number of nodes in Figure 1 may have caused confusion. We have updated Figure 1 by changing the network calculation depictions of the test statistic to a five node network for consistency. Changes to node colour have also been added for further clarification to the figure. We have also added a second series of labels to the figure for each of the steps of the null model permutation process. The first set of labels is associated with the two edge permutation null models (A/B) and the second set is associated with the cross-network edge permutation models (C/D).

Examples A/B portray the two edge permutation null models performed in this study. Examples C/D portray the two cross-network edge permutations. A/B and C/D are identical in the number of nodes and links they have, but the figures show a shuffling of links that occurs in the permutation process. The figure legend text has been update to provide further clarification. It now reads:

“Figure 1. Schematic representation of edge-based permutation models with directed network data. Four main null model steps include (i) creating a social network from the observed data, (ii) calculating a test statistic, for example, a network-level metric like degree assortativity (high-degree fishers that are coloured red primarily connect to other high-degree fishers), (iii) randomising the observation data (typically with 1000 permutations), and (iv) recording the distribution of possible test statistics. Conclusions can then be drawn by comparing the observed test statistics to the distribution test statistics, and the P-value calculated. Throughout the edge swap permutations, the fisher positions remain the same, but the configuration of edges between fishers change based on select criteria. The four null model examples shown are all used in this paper’s analysis. (A) Outgoing edge permutation allows the randomisation of all incoming links, whilst maintaining the number of nominations (outgoing links) each individual made, (B) edge permutation only allows the swap of links, by maintaining the number of nominees (incoming links) and nominations (outgoing links) each individual made in this information-sharing network. (C) Network swap permutation maintains each dyadic nomination, but randomises the networks that these nominations were made in (i.e., when individual X nominated individual Y for information sharing within three different information-sharing networks (represented by different coloured arrows), the cross-network permutation allows these three nominations to

be reassigned to any of the nine possible networks), and (D) conservative
network swap permutation maintains each dyadic nomination, but
randomises the networks that these nominations were made in, while also
controlling for the number of nominations that took place overall within
each network (i.e., when individual X nominated individual Y for
information sharing within three different information-sharing networks,
these three nominations were reassigned amongst the networks in a way
that was equal to the number of nominations in each network).”

Figure 2A: There are five depth contour lines in the figure but only four were given in the
caption (200, 1000, 3000, and 5000). I think it'd be more convenient if the depths were
provided in the figure itself.

**Response: Depth contour labels have been added to the map of Figure 2A and the**
**relevant text removed from the figure legend.**

Figure 2B: “most red” may be better with “darkest red”?

**Response: “Most red” has been changed to “darkest red” in the figure legend.**

If the size and color provide the same information, I suggest using just one (size) not both.

**Response: We thank the reviewer for their perspective on the use of node size and**
**colour change together, but we believe that changing the node size and colour is more**
**effective for showing differences in nominations with as many respondents as we have**
**in the illustrative network. Therefore, we would like to keep the illustration as it is.**

Figure 2C and 2D: Are “edge permutations” and “edge swap” the same thing? If so, I
suggest using one term consistently.

**Response: “Edge swap” has been changed to “edge permutation” throughout the**
**manuscript and supplementary information.**

I don't see any instance where the observed value is below permutations. If so, you can
remove “purple = observed values are below permutations.”

**Response: There is one instance in Figure 2D. The observed value is below the**
**permutation for the sea turtle bycatch information-sharing network.**

Figure 3: I suggest removing “purple = observed values are below permutations” because
there is none.

**Response: This was an error carried over from another figure. “purple = observed**
**values are below permutations” has been removed from the figure legend.**

Why are some permutations distributions smooth and others not?

Response: The smoothness of the permutations distributions is just in relation to the possible, or most likely, values that the permutations can generate. This is another example of why permutation analysis like this can be useful, as it directly shows what may have been expected from the underlying data itself, and makes no assumptions about normal or smooth distributions etc.

Table 2: What is the difference between filled and empty red circles, and between red and black circles?

Response: The following text has been added to the figure legend for clarification:

“Red circles and arrows highlight the relevant structure of the network each metric measures. Black and white circles and black arrows highlight structures of the network that are not relevant to the metric measure in question”

Example of degree assortativity. “The authors use simulations of animal data to assess how variation in simple social association rules between individuals can determine their positions within emerging social networks.” What kind of animal data were simulated in the study, movements?

Response: The simulated data was individuals’ association patterns. The author separately considered three simple scenarios, each with its own specified process underlying social differences between individuals. For each of these three scenarios, they carried out simulations where social associations occurred at random apart from the specified scenario to generate the arising social networks. The first scenario considered individuals’ general sociability as the number of individuals that a focal individual generally associates with (which is also analogous to gregariousness or average group size). In a second scenario, individuals were set to vary in their ‘reassociation tendency’, which was defined as their propensity to reassociate with individuals they had associated with before. In the third scenario, the authors varied individuals’ ‘within-group association’ i.e. their likelihood of associating with their own group members over non-group members.

The following text has been added to Table 2 to clarify this point.

“The authors use simulations of individual association patterns of animals to assess how variation in simple social association rules between individuals can determine their positions within emerging social networks”

Supplementary Methods

Line 97: Should “in-degree of links” be “in-going links”?

Response: “in-degree” can refer to the number in-coming links. But we have edited this sentence for clarify. The revised sentence reads:

“While the number of out-going links was limited to ten, there was no limit on the number of in-coming links in the network (i.e., there was no limit to the number of times others could nominate a skipper), which was the main focus of our analysis.”

Line 248: Is “an actor” the same as a “node”? If so, I suggest using the same term throughout and perhaps replace it with “skipper” or “fisher.”

Response: The review is correct. We have implemented the suggested change swapping “actor” for “fisher”. The relevant text reads:

“...how far a fisher is from the furthest other...”

Figure S2: There are no red or purple lines.

Response: The following text has been removed from the figure legend:

“red = observed values are above the permutations,” and “, purple = observed values are below the permutations”

Figure S3: Purple is difficult to discern. I suggest changing the color or, alternatively, use black for all - there is no information gained by adding different colors.

Response: We thank the reviewer for their perspective on the use of coloured lines in the figure, but we believe that having different coloured lines is more effective for showing key differences of whether the network metrics are above, within, or below the null distributions. We would also like to keep the colour purple. Therefore, we have kept the illustration as it is.

Table S2: Is “in assortment” the same thing as “incoming assortativity” in the main text? If so, I suggest using one term consistently throughout the manuscript, including this supplementary material. If not, the term needs to be defined somewhere. I did not find “assortment” in the main text. Similarly “variance eccentricity” was not found in the main text. Looking at other table captions, variance eccentricity should be worded as “variance in node eccentricity.” I suggest going through all figure and table captions and make sure that they are comparable.

Response: Thank you for pointing out this lack of clarity with use of terms. We have removed any reference to “assortment”, which was in error. All reference is now to “degree assortativity” in general, or “in-degree assortativity” and “out-degree assortativity” when specifically referring to incoming or outgoing ties. We have ensured consistency throughout the main manuscript and the supplementary materials.

Table S3: Here, “out assort” is defined as “assortativity coefficient for outgoing links.” Is this comparable to Table S2? If so, I suggest changing the captions so that they are comparable.

Response: The reviewer is correct. We have updated the figure legends for Tables S2 and S3 for consistency. The relevant text reads:

“Table S2. Measures of network structure with statistics describing the in-degree assortativity coefficient (in assort) and variance eccentricity (var eccent).”

“Table S3. Measures of network structure with statistics describing the out-degree assortativity coefficient (out assort), mean node eccentricity (mean eccent) and variance in node betweenness (var between).”

Once again, we thank the reviewer for their thorough and thoughtful review.

Referee: 2

Comments to the Author(s)

This paper presents an association study between multiple layers of an information sharing network, where nodes are individuals, edges represent whether information is transmitted (or not) and layers represent the type of information transmitted (vessel, regulation, where to fish, etc.). The authors assess two networks metrics, and base their analysis on similarity/differences between the different information networks comparing them to specific null models.

Two key points makes me wonder whether this is an appropriate journal for the work presented.

1) The methods are interesting, but not new, in some ways, these are very similar to the methods used to assess whether specific configurations (often called motifs) are over or under represented. ERGM (that the authors refer to in the supplementary) already take this into account (and more stringent null model as well, ERGM take into account network configurations taking into consideration (conditioned over) all other configurations/nodal characteristics chosen for a specific analysis.

2) Social networks are different than other types of network where the state of the nodes are more easily predictable or where the edges follow specific laws (entropy, energy etc.). Humans, and especially knowledge and information exchange, as well as resources and other relationship, do not necessarily adhere to general principles, and predicting is, in this context, too strong of a statement. Further, social networks can change faster than other types of networks based on programs, events, or even choices that individual decide to make. Overall, while looking at null model is interesting, this is not novel (- see point 1 - ERGM do exactly that, as well as motifs in ecological, biological and social networks). Hence the author should be clearer on where the novelty of their approach lies. As of now, this is an application to social network of edge

Arlidge et al.

29 July 2021

Royal Society Open Science – Response to Proceedings of the Royal Society B

permutation that lacks the nuances that are system specific.

Response: We thank the Reviewer for their thoughtful review of our manuscript.

The reviewer raises two interesting points. Concerning the reviewer's first key point on the novelty of the methods, we are very aware of the use of ERGMs and their extensive use throughout the sociology literature analysing social networks. However, we are only aware of two studies to the null model permutations that we use to human network data. While ERGMs can perform many of the same functions that the null models that we perform can, they are not identical and there is novelty in applying these analytical methods to human social network data.

Some other points I would like to raise:

I strongly suggest to move the Network Null model in the main manuscript and not in the supplementary, as null models have been applied in SNA... for quite some time... as you correctly state, while the impression I had reading the manuscript is that the null model idea was novel as only applied in ecology.

Response: Done. The paragraph has been moved from the supplementary material to the methods section.

Methodologically: i am have doubts in how you assess differences and similarity between the different networks. In fact, you use basically two metrics. A more comprehensive approach on network structure similarity would have been the one proposed by Przulj (2007) see also Tantardini et al: using graphlet correlation distance is considered better than other options when it comes to assess network similarity, and works even when nodes are non-matching, see Tantardini, Dimiitrova for multiplex networks, and others).

Response: We have now added a reference to the work cited here as a potential alternative method for considering differences/similarities between networks. However, we have opted to continue to use the metrics we initially used here as we believe that these are intuitive to the reader (i.e. correlation based) and easily/usefully compared to the null models that we implement here too, and allow interpretation. We think that these benefits of this approach are useful, and -while other metrics may be available- we believe that this is the best fit for our research here. Thank you for these interesting references.

Minor comments

More in detail, a couple of issues that I think, need to be addressed:

Line 183 - 185 this premise is not really justified by the references. This statement, if true (and it may be, as all structural properties of a system can have important social implications) need refinement and improvement, as well as being contextualized within the literature, preferably

related to networks and common pool resources (See some of the work of Bodin and Crona in this area)

On homophily: this is a bit far fetched, it would be true if and only if you can assess specific individual characteristics that go beyond the structural characteristics you can assess by looking at assortativity based on in/out degree. In fact, yes, we tend to share information based on similarity, but that similarity is often related to background, kinship, system representation etc.. If you go this route, you should also assess literature on prestige bias imitation.

Response: We have refined the sentence and its referencing:

“The degree assortativity (or homophily) coefficient measures the extent to which central fishers are connected to other central fishers, and peripheral fishers are connected to other peripheral fishers based on a particular trait [64, 65]. The level of degree assortativity in a network is known to have important social implications for the operation and emergence of competition and cooperation (e.g., small-scale fishers working with the same gear type will form social relationships involving information exchange [30, 66]).”

The added references include:

Alexander S., Bodin Ö., Barnes M. 2018 Untangling the drivers of community cohesion in small-scale fisheries. *Int J Commons* 12(1), 519–547.

Crona B., Bodin Ö. 2006 What you know is who you know? Communication patterns among resource users as a prerequisite for co-management. *Ecol Soc* 11(2).

Line 186: this implies perfect correlation not homophily.

Response: We have amended the sentence to reflect this point. The sentence now reads:

“Positive values demonstrate degree assortativity, whereby a score of 1 would indicate that the network is assorted by individuals’ degree to the maximum extent, and negative values represent disassortment (i.e. central individuals more likely to be associated with peripheral individuals).”

Line 241: You state that you can predict a specific network multiple times, I am unsure whether I mis-interpreted, but I challenge the notion that you are looking at "predicting". At best you are looking at association in a specific point in time. Human systems are a bit more difficult to analyze and predict than any other systems. Again, you should change the language accordingly and talk mainly about association/correlation.

Response: We have amended the sentence using the term “correlated” rather than “predicted” as the reviewer suggest. The sentence now reads:

**“To reveal the extent to which the sea turtle bycatch information-sharing**
**network can be correlated with the other networks evaluated, we examined the**
**dyadic similarity between the different information-sharing networks.”**

**We also amended a sentence in the conclusion:**

**“Our study demonstrates how networks of information-sharing regarding a**
**conservation-relevant topic (sea turtle bycatch) are structurally dissimilar**
**from other types of fishing information-sharing, and the extent to which**
**fisher-fisher (dyadic) ties can be correlated with other information-sharing**
**networks”**

Use of Mantel test: Describing the use of your correlation between networks as a mantel
test brings issues, as it has been shown (see Legendre et al.) that it is not a good choice
for data that are related. Network data are by definition not-independent.

**Response: Thanks for this comment. Mantel tests are commonly used as a simple**
**descriptor of the correlation between network matrices, and provide an intuitive measure**
**of this. As stated in the responses above, we now acknowledge that other measures of graph**
**similarity are available, but we have opted to continue to use our initial correlation**
**measure here (for the reasons provided above). Further, the combination of using this**
**correlation metric on the directed network matrices with the comparison to the null models**
**means that the ‘related’ data issues are directly considered within the analyses anyway (as**
**described in the Methods and Supplementary information). Therefore, thank the reviewer**
**for their comment, but have opted to keep this method, and have added references to other**
**potential methods for the interested reader to refer to if they wish.**

Even more so, it is likely that sharing information in one network leads to sharing in all other
networks. This may also be an artifact (to some extent) on how the questionnaire was laid out
(name and then the different networks, as shown in the supplementary material).

**Response: Respondents were interviewed verbally and did not see the recording sheet that**
**is presented in supplementary materials. This was not explicitly stated in the**
**supplementary material, so it has been added”**

**“All respondents were interviewed verbally using a face-to-face interview**
**format.”**

**And supplementary material lines:**

**“Respondents were not shown the reporting table used by the interviewers to**
**records each respondent’s responses (Q25 of the social network**
**questionnaire).”**

**For further clarification of how the data was gathered, the prompt presented in full in the**
**supplementary information was given to the respondents before the fixed response:**

“Respondents were asked to consider people from San Jose that they share useful information about fishing with; considering those that they thought may influence their fishing success. Respondents were reminded that the shared information and names will remain anonymous and will not be revealed. We highlighted that the information provided will help us understand how information that relates to fishing flows between fishers. Prior to the fixed response, respondents were asked to consider relationships that they have had with other vessel owners, captains, owner/captains (owners who also captain their vessel), other fishery leaders, fishery management officials, members of the scientific or not-for-profit community, boat launching / landing support, fish transport associations, fish sellers/market operators, their family and friends, and any other people they have fished with, or shared information with about fishing over the last 5 years.”

Line 249: unfolded matrices: not sure what this means

Response: We have now removed the term ‘unfolded matrices’.

some References in case the authors want to incorporate some of the suggestions made:

Response: We thank the Reviewer for including this reference list. This additional reading was informative and we incorporated Pržulj, N. (2007) and Tantardini et al. (2019) into our reference list.

Dimitrova, T., Petrovski, K., & Kocarev, L. (2020). Graphlets in Multiplex Networks. *Scientific Reports*, 10(1), 1–13. <https://doi.org/10.1038/s41598-020-57609-3>

Legendre, P., Fortin, M. J., & Borcard, D. (2015). Should the Mantel test be used in spatial analysis?. *Methods in Ecology and Evolution*, 6(11), 1239-1247.

Pržulj, N. (2007). Biological network comparison using graphlet degree distribution. *Bioinformatics*, 23(2), 177–183. <https://doi.org/10.1093/bioinformatics/btl301>

Shanafelt, D. W., Salau, K. R., & Baggio, J. A. (2017). Do-it-yourself networks: a novel method of generating weighted networks. *Royal Society open science*, 4(11), 171227.

Tantardini, M., Ieva, F., Tajoli, L., & Piccardi, C. (2019). Comparing methods for comparing networks. *Scientific Reports*, 9(1), 1–19. <https://doi.org/10.1038/s41598-019-53708-y>

Yaveroaa Lu, Ö. N., Malod-Dognin, N., Davis, D., Levnajic, Z., Janjic, V., Karapandza, R., ... Pržulj, N. (2014). Revealing the hidden Language of complex networks. *Scientific Reports*, 4, 1–9. <https://doi.org/10.1038/srep04547>

Assessing information-sharing networks within small-scale fisheries and the implications for conservation interventions

William N. S. Arlidge^{1,2,3*}, Josh A. Firth^{4,5}, Joanna Alfaro-Shigueto^{6,7,8},
Bruno Ibanez-Erquiaga^{9,10}, Jeffrey C. Mangel^{6,7}, Dale Squires¹¹, E.J.
Milner-Gulland¹

¹ Department of Biology and Ecology of Fishes, Leibniz-Institute of Freshwater Ecology and Inland Fisheries, Müggelseedamm 310, 12587 Berlin, Germany

² Faculty of Life Sciences, Humboldt-Universität zu Berlin, Invalidenstrasse 42, 10115 Berlin, Germany

³ Interdisciplinary Centre for Conservation Science, Department of Zoology, University of Oxford OX1 3PS, UK

⁴ Edward Grey Institute, Department of Zoology, University of Oxford, Oxford OX1 3PS, UK

⁵ Merton College, University of Oxford, Oxford, UK

⁶ Pro Delphinus, Calle José Galvez 780-E, Miraflores 15074, Perú

⁷ School of Biosciences, University of Exeter, Cornwall Campus, Penryn, Cornwall, TR10 9FE, UK

⁸ Facultad de Biología Marina, Universidad Científica del Sur, Campus Villa, Lima 42, Perú

⁹ Departamento de Química y Biología, Universidad San Ignacio de Loyola, Lima, Perú

¹⁰ Asociación CONSERVACCION, Lima, Peru.

¹¹ NOAA Fisheries Southwest Fisheries Science Center, La Jolla, CA, USA

[revised manuscript text omitted]

In this study, we apply a social network analysis and permutation-based null model approach to assess whether networks of small-scale fisher's information sharing about sea turtle bycatch are structurally similar to other types of fishing information-sharing networks (Table 1). We test the assumption that knowledge about information-sharing social networks should be transferable to a related information-sharing network of interest (other fishing issues and sea turtle bycatch, in our case). We illustrate how null model analysis techniques used in the ecological sciences may offer more in-depth insights into the fine-scale structure of human networks than could be gained from simple centrality measurement methods, and we provide insight into comparing information-sharing networks within a social system of high conservation interest. We conclude by discussing how our findings can contribute to understanding how information related to conservation interventions may spread socially.

3. Materials and Methods

Study system

During our survey period of 1 July – 30 September 2017, San Jose, Lambayeque, Peru (6°46' S, 79°58' W) was home to 168 small-scale commercial gillnet skippers that fish throughout the year. We surveyed 165 fishers representing 98.2% of the gillnet skippers at the site (Fig. S2b, Table S1). Gillnet skippers in San Jose are known to capture sea turtles in high numbers [16, 40, 45]. Green turtles (*Chelonia mydas*) are captured most frequently, followed by olive ridley turtles (*Lepidochelys olivacea*), and leatherback turtles (*Dermochelys coriacea*) [42]. At the time of the study, five gillnet skippers and their crew were involved in a trial community co-management bycatch reduction scheme operating from San Jose that requires fishers to use light-emitting diodes on their nets to reduce sea turtle bycatch [43]. Skippers were deemed active if they fished from the San Jose port with gillnets in the winter of 1 July – 30 September 2017. The network was surveyed during winter as skippers actively fishing during these months are established fishers in the San Jose community throughout the year. We define gillnets as encompassing surface drift gillnets and fixed bottom gillnets in single or trammel net configurations. The total San Jose gillnet skipper population (n=168) was determined using a combination of membership lists of the two main fishing groups in San Jose, lists of boats towed in and out of the water with tractors, and key informant interviews (Supplementary Information).

Data collection

Detailed social network data was collected using a structured questionnaire with a fixed choice survey design. Respondents were asked to consider up to ten individuals with whom they exchange useful information about fishing and whom they considered valuable to their fishing success. We classified nine fine-scale information-sharing types about which we expect gillnet skippers to exchange fishing related information (Table 1). As each nominee was given by the respondents, they were asked to highlight which fishing information type they discussed with each nominee. For each fishing information type, respondents were asked to consider relationships that they have had with other skippers, vessel owners, crew members, other fishery leaders, fishery management officials, members of the scientific community, boat launching/landing support, fish sellers/market operators, family members, and any other stakeholders they fished or shared information with about fishing. Respondents were not asked who they receive information from. Interviews were undertaken verbally and respondents were not shown the questionnaire where responses were written (Supplementary Information). Questionnaires were trialed with fishers (n=8) in the Santa Rosa fishing community 17 km down

the coast from San Jose (Fig. 2a). Pilot study data were not included in this study's analysis. Fishers were interviewed in their native language (Spanish).

Statistics and Reproducibility

Social network construction

A social network was created for each fishing information type (Table 1). In each network, the nodes were the fishers, and the binary directed edges were the nominations by one fisher (sender) of another fisher (receiver) for this information type. All analysis was carried out in R [46] using the igraph package [47] for visualising and processing the analysis and carrying out the network comparisons using the null models.

Structural differences across information-sharing networks

To investigate whether networks of information-sharing between individuals were similar across different information types, we examined the networks' structural properties in terms of their **degree assortativity and the variance and mean of individual centrality** (Table 2). To account for the effect of basic characteristics of the networks (e.g., number of ties, degree distributions) we compared these observed summary statistics to null models, which allowed inference of structural differences and similarities over and above that expected from these simple differences using null models (Fig. 1).

Network null models (routines that generate different types of null datasets against which the observed dataset can be compared) are a group of statistical models commonly applied in network analysis. Specifically, null models are especially useful when investigating hypotheses in datasets, control groups are difficult to establish, exogenous treatments are unavailable, and observations may be missing or biased [48-50]. As such, null model methods are important because network data is comprised of non-independent observations of multiple individuals, and small variations in how data are collected between respondents can easily generate patterns that appear as social structure [50, 51]. Null models have been applied to network data in sociology since the 1970s [48] and discipline-specific developments have subsequently been made to statistical models such as exponential random graph models [52, 53], conditional uniform graph tests [54-56] and quadratic assignment procedure tests [57-59]. Since the mid-1990s, the field of ecology has also made extensive use of null models to develop specialised hypothesis testing routines and treat underlying uncertainty or data collection methodology biases when interrogating non-human animal network data [60-62]. **Here we expand the application of the permutation-based null model approach routinely used in ecology to human social networks, which have also been applied in the field of epidemiology for assessing human contact tracing disease control measures [63].**

Degree assortativity

The degree assortativity (or **homophily**) coefficient measures the extent to which central fishers are connected to other central fishers, and peripheral fishers are connected to other peripheral fishers based on a particular trait [64, 65]. The level of degree assortativity in a network is known to have important social implications for the operation and emergence of competition and cooperation (e.g., small-scale fishers working with the same gear type will form social relationships involving information exchange [30, 66]). Positive values demonstrate degree assortativity, whereby a score of 1 would indicate that the network is assorted by individuals' degree to the maximum extent, and negative values represent disassortment (i.e., central individuals more likely to be associated with peripheral individuals). When fishers of similar centrality are randomly distributed in a community (i.e., disassorted), those networks do not always score -1 due to the minimum value depending on the number of fishers and the relative number of ties within each group [64]. For each of the information-sharing networks, we first calculated the assortativity by in-degree (the number of nominations each interviewed fisher received). Degree assortativity is the primary assortativity measure of interest as in-degree provides the measure of which fishers provide information to others. However, as fishers differed in the number of nominations they made for each information-sharing type, we also calculated the assortativity by out-degree (the number of nominations each fisher made) to examine whether fishers were also disproportionately connected to others who make a similar number of nominations as themselves. As social networks often show assortativity by degree, we predicted that all the information sharing networks would be positively homophilous by nominations made and nominations received (i.e., highly nominating and nominated fishers would be closely associated with highly nominating and nominated fishers, whilst peripheral fishers would be more likely to be connected).

Eccentricity

We aimed to consider node-level properties that depend on the structure of the social network (Table 2). 
[revised manuscript text omitted]

- Wright A.J., Verissimo D., Pilfold K., Parsons E., Ventre K., Cousins J., Jefferson R., Koldewey H., Llewellyn F., McKinley E. 2015 Competitive outreach in the 21st century: Why we need conservation marketing. *Ocean Coast Manage* **115**, 41-48.
- Gutiérrez N.L., Hilborn R., Defeo O. 2011 Leadership, social capital and incentives promote successful fisheries. *Nature* **470**(7334), 386.
- Heimlich J.E. 2010 Environmental education evaluation: Reinterpreting education as a strategy for meeting mission. *Evaluation and Program planning* **33**(2), 180-185.
- Groce J.E., Farrelly M.A., Jorgensen B.S., Cook C.N. 2019 Using social-network research to improve outcomes in natural resource management. *Conserv Biol* **33**(1), 53-65. (doi:doi:10.1111/cobi.13127).
- de Lange E., Milner-Gulland E.J., Keane A. 2019 Improving Environmental Interventions by Understanding Information Flows. *Trends Ecol Evol* **34**(11), 1034-1047. (doi:<https://doi.org/10.1016/j.tree.2019.06.007>).
- Davies R., Cripps S., Nickson A., Porter G. 2009 Defining and estimating global marine fisheries bycatch. *Mar Policy* **33**(4), 661-672.
- Kennelly S.J. 2020 Bycatch Beknown: Methodology for jurisdictional reporting of fisheries discards—Using Australia as a case study. *Fish Fish* **21**(5), 1046-1066.
- Hall M.A. 1996 On bycatches. *Rev Fish Biol Fish* **6**(3), 319-352.
- Lewison R.L., Crowder L.B., Wallace B.P., Moore J.E., Cox T., Zydalis R., McDonald S., DiMatteo A., Dunn D.C., Kot C.Y. 2014 Global patterns of marine mammal, seabird, and sea turtle bycatch reveal taxa-specific and cumulative megafauna hotspots. *Proc Natl Acad Sci* **111**(14), 5271-5276.
- Gray C.A., Kennelly S.J. 2018 Bycatches of endangered, threatened and protected species in marine fisheries. *Rev Fish Biol Fish* **28**(3), 521-541.
- Berkes F., Mahon R., McConney P., Pollnac R., Pomeroy R. 2001 *Managing Small-Scale Fisheries: Alternative Directions and Methods*, International Development Research Centre; 309 p.
- Arthur R.I. 2020 Small-scale fisheries management and the problem of open access. *Mar Policy*, 103867.
- FAO. 2015 Voluntary Guidelines for Securing Sustainable Small-Scale Fisheries. (Rome, 2015, FAO).
- Alfaro-Shigueto J., Mangel J.C., Darquea J., Donoso M., Baquero A., Doherty P.D., Godley B.J. 2018 Untangling the impacts of nets in the southeastern Pacific: Rapid assessment of marine turtle bycatch to set conservation priorities in small-scale fisheries. *Fish Res* **206**, 185-192. (doi:<https://doi.org/10.1016/j.fishres.2018.04.013>).
- Temple A.J., Wambiji N., Poonian C.N., Jiddawi N., Stead S.M., Kiszka J.J., Berggren P. 2019 Marine megafauna catch in southwestern Indian Ocean small-scale fisheries from landings data. *Biol Conserv* **230**, 113-121.
- Peckham S.H., Diaz D.M., Walli A., Ruiz G., Crowder L.B., Nichols W.J. 2007 Small-scale fisheries bycatch jeopardizes endangered Pacific loggerhead turtles. *PLoS ONE* **2**(10), e1041.
- Chuenpagdee R., Jentoft S. 2019 Small-Scale Fisheries: Too Important to Fail. In *The Future of Ocean Governance and Capacity Development* (pp. 349-353, Brill Nijhoff).
- Ostrom E. 1990 *Governing the commons: The evolution of institutions for collective action*, Cambridge university press.
- Alexander S.M., Staniczenko P.P.A., Bodin Ö. 2020 Social ties explain catch portfolios of small-scale fishers in the Caribbean. *Fish Fish* **21**(1), 120-131. (doi:10.1111/faf.12421).
- Friedkin N.E., Johnsen E.C. 1997 Social positions in influence networks. *Social Networks* **19**(3), 209-222. (doi:[https://doi.org/10.1016/S0378-8733\(96\)00298-5](https://doi.org/10.1016/S0378-8733(96)00298-5)).
- Abrahamse W., Steg L. 2013 Social influence approaches to encourage resource conservation: A meta-analysis. *Global Environ Change* **23**(6), 1773-1785.
- Dodds P.S., Watts D.J. 2004 Universal Behavior in a Generalized Model of Contagion. *Physical Review Letters* **92**(21), 218701. (doi:10.1103/PhysRevLett.92.218701).
- Centola D. 2018 *How Behavior Spreads: The Science of Complex Contagions*, Princeton University Press.

26. Zhang A.J., Matous P., Tan D.K.Y. 2020 Forget opinion leaders: the role of social network brokers in the adoption of innovative farming practices in North-western Cambodia. *International Journal of Agricultural Sustainability*, 1-19. (doi:10.1080/14735903.2020.1769808).
27. Pornpitakpan C. 2004 The persuasiveness of source credibility: A critical review of five decades' evidence. *Journal of applied social psychology* **34**(2), 243-281.
28. McDonald R.I., Crandall C.S. 2015 Social norms and social influence. *Curr Opin Behav Sci* **3**, 147-151.
29. Turner R.A., Polunin N.V., Stead S.M. 2014 Social networks and fishers' behavior: exploring the links between information flow and fishing success in the Northumberland lobster fishery. *Ecol Soc* **19**(2).
30. Alexander S., Bodin Ö., Barnes M. 2018 Untangling the drivers of community cohesion in small-scale fisheries. *Int J Commons* **12**(1), 519-547.
31. Stevens K., Frank K.A., Kramer D.B. 2015 Do social networks influence small-scale fishermen's enforcement of sea tenure? *PLoS ONE* **10**(3), e0121431.
32. Barnes M.L., Bodin Ö., McClanahan T.R., Kittinger J.N., Hoey A.S., Gaoue O.G., Graham N.A. 2019 Social-ecological alignment and ecological conditions in coral reefs. *Nat Comm* **10**(1), 1-10.
33. Bodin Ö., García M.M., Robins G. 2020 Reconciling Conflict and Cooperation in Environmental Governance: A Social Network Perspective. *Annu Rev Environ Res* **45**(1), 471-495. (doi:10.1146/annurev-environ-011020-064352).
34. Barnes-Mauthe M., Arita S., Allen S., Gray S., Leung P. 2013 The influence of ethnic diversity on social network structure in a common-pool resource system: implications for collaborative management. *Ecol Soc* **18**(1).
35. Alexander S.M., Armitage D., Charles A. 2015 Social networks and transitions to co-management in Jamaican marine reserves and small-scale fisheries. *Global Environ Change* **35**, 213-225.
36. Mbaru E.K., Barnes M.L. 2017 Key players in conservation diffusion: Using social network analysis to identify critical injection points. *Biol Conserv* **210**, 222-232.
37. Barnes M.L., Lynham J., Kalberg K., Leung P. 2016 Social networks and environmental outcomes. *Proc Natl Acad Sci* **113**(23), 6466-6471.
38. Dunbar R. 2018 The anatomy of friendship. *Trends in cognitive sciences* **22**(1), 32-51.
39. Barrett L., Henzi S.P., Lusseau D. 2012 Taking sociality seriously: the structure of multi-dimensional social networks as a source of information for individuals. *Philosophical Transactions of the Royal Society B: Biological Sciences* **367**(1599), 2108-2118.
40. Alfaro-Shigueto J., Dutton P.H., Van Bresse M., Mangel J. 2007 Interactions between leatherback turtles and Peruvian artisanal fisheries. *Chelonian Conserv Biol* **6**(1), 129-134.
41. Alfaro-Shigueto J., Mangel J.C., Bernedo F., Dutton P.H., Seminoff J.A., Godley B.J. 2011 Small-scale fisheries of Peru: a major sink for marine turtles in the Pacific. *J Appl Ecol* **48**(6), 1432-1440.
42. Arlidge W.N.S., Squires D., Alfaro-Shigueto J., Booth H., Mangel J.C., Milner-Gulland E.J. 2020 A Mitigation Hierarchy Approach for Managing Sea Turtle Captures in Small-Scale Fisheries. *Front Mar Sci* **7**(49). (doi:10.3389/fmars.2020.00049).
43. Ortiz N., Mangel J.C., Wang J., Alfaro-Shigueto J., Pingo S., Jimenez A., Suarez T., Swimmer Y., Carvalho F., Godley B. 2016 Reducing green turtle bycatch in small-scale fisheries using illuminated gillnets: The Cost of Saving a Sea Turtle. *Marine Ecological Progress Series* **545**, 251-259.
44. Alfaro-Shigueto J., Mangel J.C., Dutton P.H., Seminoff J.A., Godley B.J. 2012 Trading information for conservation: a novel use of radio broadcasting to reduce sea turtle bycatch. *Oryx* **46**(03), 332-339.
45. Alfaro-Shigueto J., Mangel J.C., Pajuelo M., Dutton P.H., Seminoff J.A., Godley B.J. 2010 Where small can have a large impact: structure and characterization of small-scale fisheries in Peru. *Fish Res* **106**(1), 8-17.
46. R Core Team. 2019 R: A language and environment for statistical computing. (Vienna, Austria: R Foundation for Statistical Computing).
47. Csardi G., Nepusz T. 2006 The igraph software package for complex network research. *InterJournal, Complex Systems* **1695**(5), 1-9.
48. Davis J.A. 1970 Clustering and hierarchy in interpersonal relations: Testing two graph theoretical models on 742 sociomatrices. *American Sociological Review*, 843-851.
49. Gotelli N., Graves G. 1996 *Null models in ecology* Washington, DC, USA, Smithsonian Institution Press.
50. Whitehead H. 2008 *Analyzing animal societies: quantitative methods for vertebrate social analysis*, University of Chicago Press.
51. Farine D.R. 2017 A guide to null models for animal social network analysis. *Methods Ecol Evo* **8**(10), 1309-1320.
52. Snijders T.A. 2002 Markov chain Monte Carlo estimation of exponential random graph models. *Journal of Social Structure* **3**(2), 1-40.
53. Snijders T.A., Pattison P.E., Robins G.L., Handcock M.S. 2006 New specifications for exponential random graph models. *J Sociological methodology* **36**(1), 99-153.
54. Katz L., Wilson T.R. 1956 The variance of the number of mutual choices in sociometry. *Psychometrika* **21**(3), 299-304.
55. Holland P.W., Leinhardt S. 1977 A method for detecting structure in sociometric data. In *Social Networks* (pp. 411-432, Elsevier).
56. Anderson B.S., Butts C., Carley K. 1999 The interaction of size and density with graph-level indices. *Social Networks* **21**(3), 239-267.
57. Hubert L. 1986 *Assignment methods in combinational data analysis*, CRC Press.
58. Krackardt D. 1987 QAP partialling as a test of spuriousness. *Social Networks* **9**(2), 171-186.
59. Dekker D., Krackhardt D., Snijders T.A. 2007 Sensitivity of MRQAP tests to collinearity and autocorrelation conditions. *Psychometrika* **72**(4), 563-581.
60. Bejder L., Fletcher D., Bräger S. 1998 A method for testing association patterns of social animals. *Anim Behav* **56**(3), 719-725.
61. Croft D.P., James R., Krause J. 2008 *Exploring animal social networks*, Princeton University Press.
62. Whitehead H. 1995 Investigating structure and temporal scale in social organizations using identified individuals. *Behav Ecol* **6**(2), 199-208.
63. Firth J.A., Hellewell J., Klepac P., Kissler S., Jit M., Atkins K.E., Clifford S., Villabona-Arenas C.J., Meakin S.R., Diamond C., et al. 2020 Using a real-world network to model localized COVID-19 control strategies. *Nat Med* **26**(10), 1616-1622. (doi:10.1038/s41591-020-1036-8).
64. Newman M.E. 2003 Mixing patterns in networks. *Physical Review E* **67**(2), 026126.
65. McPherson M., Smith-Lovin L., Cook J.M. 2001 Birds of a Feather: Homophily in Social Networks. *Annu Rev Socio* **27**(1), 415-444. (doi:10.1146/annurev.soc.27.1.415).
66. Crona B., Bodin Ö. 2006 What you know is who you know? Communication patterns among resource users as a prerequisite for co-management. *Ecol Soc* **11**(2).
67. Hage P., Harary F. 1995 Eccentricity and centrality in networks. *Social Networks* **17**(1), 57-63.
68. Pržulj N. 2007 Biological network comparison using graphlet degree distribution. *Bioinformatics* **23**(2), e177-e183. (doi:10.1093/bioinformatics/btl301).
69. Tantardini M., Ieva F., Tajoli L., Piccardi C. 2019 Comparing methods for comparing networks. *Scientific Reports* **9**(1), 17557. (doi:10.1038/s41598-019-53708-y).
70. Mantel N. 1967 The detection of disease clustering and a generalized regression approach. *Cancer Res* **27**(2 Part 1), 209-220.
71. Newman M.E. 2002 Assortative mixing in networks. *Physical review letters* **89**(20), 208701.
72. Watts D.J., Strogatz S.H. 1998 Collective dynamics of 'small-world' networks. *Nature* **393**(6684), 440.
73. Albert R., Barabási A.-L. 2002 Statistical mechanics of complex networks. *Reviews of Modern Physics* **74**(1), 47-97. (doi:10.1103/RevModPhys.74.47).
74. Campbell E., Salathé M. 2013 Complex social contagion makes networks more vulnerable to disease outbreaks. *Scientific reports* **3**(1), 1-6.
75. Zhang J., Centola D. 2019 Social networks and health: new developments in

diffusion, online and offline. *Annu Rev Socio*
45, 91-109.

76. Fink C., Schmidt A., Barash V.,
Cameron C., Macy M. 2016 Complex
contagions and the diffusion of popular Twitter
hashtags in Nigeria. *Social Network Analysis
and Mining* **6**(1), 1.

77. Montanari A., Saberi A. 2010 The
spread of innovations in social networks. *Proc
Natl Acad Sci* **107**(47), 20196-20201.

78. Firth J.A., Albery G.F., Beck K.B.,
Jarić I., Spurgin L.G., Sheldon B.C., Hoppitt W.
2020 Analysing the Social Spread of Behaviour:
Integrating Complex Contagions into Network
Based Diffusions. *arXiv preprint
arXiv:201208925*.

79. Marsden P.V., Friedkin N.E. 1993
Network studies of social influence.
Sociological Methods & Research **22**(1), 127-
151.

80. Matous P., Wang P. 2019 External
exposure, boundary-spanning, and opinion
leadership in remote communities: A network
experiment. *Social Networks* **56**, 10-22.

81. Firth J.A., Sheldon B.C., Brent L.J.
2017 Indirectly connected: simple social
differences can explain the causes and
apparent consequences of complex social
network positions. *Proc R Soc B* **284**(1867),
20171939.

82. Corlew L.K., Keener V., Finucane
M., Brewington L., Nunn-Crichton R. 2015
Using social network analysis to assess
communications and develop networking tools
among climate change professionals across the
Pacific Islands region. *Psychosocial Intervention*
24(3), 133-146.
(doi:<https://doi.org/10.1016/j.psi.2015.07.004>)

.

R. Soc. open sci. article

13

Tables**Table 1.** Information-sharing networks that relate to fishing.

Full name	Short name	Description	Broad categorisation
Sea turtle bycatch	T.Bycatch	Sea turtle bycatch encounters including live releases and mortalities in nets.	Process of fishing, Business and governance of fishing
Gillnet type & maintenance	Gear	Changes made to net configuration (shifting rigging configurations from surface drift net to mid-water drift net or bottom-set net), and net maintenance.	
Weather conditions	Weather	Ocean and weather conditions (e.g., wind, swell).	
Fish location & catch sites	Location	Where fish might be located and where they have been travelling to fish.	Process of fishing
Fishing activity	Activity	How many people fishing, who is fishing, who caught what.	
Vessel technology & maintenance	Tech	Existing and new technologies used onboard the vessel (e.g., echo sounder, compass) and vessel maintenance (e.g., hull repairs, painting).	
Fishing regulations	Regs	Fishery policy and legislation.	
Fishing finances	Finance	Market prices, loans, fines, penalties.	Business and governance of fishing
Crew management	Crew	The hiring and instructing of crew onboard the vessel.	

Table 2. Network metrics used to assess information-sharing network structure. Fishers (circles) and ties (arrows) outline the represented metric in the network. Red circles and arrows highlight the relevant structure of the network each metric measures. Black and white circles and black arrows highlight structures of the network that are not relevant to the metric measure in question.

Metric	Network structure	Definition	Theoretical use in conservation-relevant systems	Example
Degree assortativity (homophily)		A preference for individuals to attach to others that are similar in some way (e.g., high-degree) [64]	Identifies individuals and pathways of individuals that could facilitate widespread diffusion of information about conservation initiatives in a community of conservation interest.	The authors use simulations of individual association patterns of animals to assess how variation in simple social association rules between individuals can determine their positions within emerging social networks. The results show that simple differences in group size cause positive assortativity and that metrics of individuals' indirect links can be more strongly related to underlying simple social differences than metrics of their dyadic links[81].

[revised manuscript text omitted]

Appendix B

Associate Editor Comments to Author:

The two reviewers have offered a range of comments that you should address in this revision. While one of the reviewers expresses concerns regarding the utility/value of the approaches adopted, if you can persuade the editors and reviewers that your paper has archival value here (rather than perhaps being a paradigm-shifting approach), the journal would be able to accept it for publication: we do not require ground-breaking novelty, but there should be some purpose to the work if it is to be published as archivally useful. Good luck with the revisions, and we'll look forward to receiving these in due course. All best.

Author response

We thank the Associate Editor for the opportunity to resubmit our manuscript. We have addressed both reviewers' comments in detail below and provide responses to their questions in **grey** below.

We believe that our approach is, as the Reviewer 1 highlights, useful as a coarse grained assessment of whether certain network metrics are more or less expected than by chance. Furthermore, while we do not believe the research presented in this manuscript has ground-breaking novelty in our application of our specific permutation methodology, we do note that we expand on the application of these specific permutation analyses beyond the original field of application, which has benefit in demonstrating the flexibility and applicability of the analytical method in question to other areas of research.

Furthermore, we believe there is utility in the results presented and that this manuscript has both theoretical and applied contributions that are valid using our applied methodology. We now highlight in the introduction that the comparison of network structures allowed us to explore whether it would be possible to design an effective (if sub-optimal) bycatch intervention by identifying and targeting influential individuals in a network sharing information on other related topics. On the applied side, this is a pertinent question because information-sharing about sea turtle bycatch might be sensitive and therefore hard to quantify, or it may be that the information-sharing networks for other topics are already known so the cost of describing a sea turtle bycatch-relevant network would not need to be incurred. It is also interesting more generally to explore how the networks for sharing different types of fisheries information resemble each other, as this may give insights into how different kinds of information are perceived. Theoretically, as we highlight in our comments to reviewer's below, we believe this study is a timely word of warning as social network analysis is more widely applied in conservation science and natural resource management (Groce et al. 2019). We have seen more than one high-quality study in the conservation science literature that has mapped general 'fishing information' networks and then drawn more specific conclusions for conservation outcomes from that data. For example, in an exemplar and important study (Barnes et al. 2016), it was intuitively assumed that information shared between fishers about fishing would be predictive of a finer-scale yet closely related

environmental outcome – shark bycatch. Similarly, in a contemporary study (Mbaru & Barnes 2017) investigating how ‘key players’ were positioned implementing broad conservation objectives, the social networks were based on similar information-sharing data mapping whom respondents fished with or exchange information about fishing. Therefore, conservation science researchers and practitioners should carefully consider how the topic of information mapped to a network relates to the desired conservation intervention when predicting environmental outcomes.

Reviewer comments to Author:

Reviewer: 1

Comments to the Author(s)

The author have replied to my previous queries.

We thank the Reviewer for their review and especially for taking the time to review our manuscript for a second time.

I still have serious doubts on assessing similarity of networks via the Mantell Test, I think that the use of graphlets as described in Przuji 2007 and Tantardini et al. 2019 would be a better approach. Except this, I also think that the permutation exercise is interesting but lacks the nuances of more "constraining" null models that maintain, for example, the number of triangles in the network. Still it can work as a coarse grained assessment of whether certain network metrics are more or less expected than by chance.

We agree with the Reviewer that use of graphlets for comparing networks could provide a finer-scale understanding of network differences than the permutation approach we have employed in the current study. However, we believe that our approach is, as the Reviewer highlights, useful as a coarse grained assessment of whether certain network metrics are more or less expected than by chance.

I have no other comments, but two minor points that relate to language:

Both on page 21 (out of 36)

Lines: 47- 48: i think the authors need to be more clear. In fact, randomness should be represented by an assortativity coefficient of 0. Disassortative networks are the ones in which high degree nodes tend to connect to low degree nodes... and assortative networks the ones in which high degree nodes tend to connect to other high degree nodes.

We have clarified this section of text, which now reads:

“A degree assortativity coefficient of zero represents randomness. Positive values demonstrate degree assortativity in which high degree nodes tend to connect to other high degree nodes, whereby a score of 1 would indicate that the network is assorted by individuals’ degree to the maximum extent. Negative values represent disassortment (i.e., high degree individuals are more likely to be associated with low degree individuals). When fishers of similar centrality are disassorted in a community, those networks do not always score -1 because the minimum value depends on the number of fishers and the relative number of ties within each group (Newman 2003).”

Line 60: something is missing here... i think it should read: peripheral fishers would be more likely to be connected to other peripheral fishers.

We have corrected this error. “... to other peripheral fishers” was missing from the end of the sentence.

Once again, we thank the Reviewer for their constructive critique of our study and for taking the time to review our manuscript for a second time at RSOS.

Reviewer: 2

Comments to the Author(s)

This manuscript assesses the information-sharing network in a small-scale fishery. The authors build different information-sharing networks based on the type of information that is shared by fishers (turtle bycatch, where to fish, regulation, etc.), and compare their structure using permutation-based null models and two key metrics: degree assortativity and node eccentricity. The authors use their results to highlight how differences in the structure of how different types of information that is shared between fishers, have implications for the diffusion of conservation interventions. The article is well written and succinct and the topic would be of interest to this journal’s audience.

We thank the Reviewer for their thorough and informative review, which has supported us in a clearer and more robust presentation of our research.

There are two main concerns I have with this paper. The first concern is that it is not clear why the measures selected (degree assortativity and node eccentricity) are appropriate to address the research question in the particular context. They cite a couple of studies of small scale fisheries to support their claim that degree assortativity is “known to have important social implications for the operation and emergence of competition and cooperation”, but these two studies actually assess homophily based on actor attribute (e.g. gear type used), which is very different to the type of homophily effect captured by degree assortativity (which is based on the centrality of the actor).

Thanks for highlighting the incongruence between the statement in text and the cited literature. We have removed the two papers previously cited (Alexander et al. 2018 and

Crona and Bodin 2006) and we have clarified our justification for using degree assortativity and node eccentricity as our primary measures of network structure in the text.

The relevant text added for degree assortativity includes the following paragraph in the methods section:

“The assortativity coefficient (akin to homophily (McPherson et al. 2001)) measures the extent to which central fishers are connected to other central fishers, and peripheral fishers are connected to other peripheral fishers based on a particular trait (Newman 2002; Newman 2003; Newman & Park 2003). The level of degree assortativity in a network is known to have important social implications for information transfer, and for the operation and emergence of competition and cooperation (McPherson et al. 2001; Newman 2003). Degree assortativity can, for example, influence the potential for a social contagion to spread, given its starting point (Centola 2011, 2018). To inform the planned expansion of the sea turtle bycatch reduction initiative in our study system, we were interested in understanding the general structure of multiple subtopics of fishing-related information that relate to the intervention and how the information-sharing networks relate to one another. Moreover, evaluating who talks to whom (i.e., directed network ties) has implications for how information may or may not flow. This is because individuals within a network can be highly central (generally nominated by many others) but just receive information – resulting in knowledge accumulation and the impeding rather than facilitation of information flow (Weiss et al. 2012; Zhang et al. 2020). Therefore, degree assortativity was the primary assortativity measure of interest as degree provides a measure of which fishers provide information to others (in-degree) and receive information from others (out-degree).”

For node eccentricity:

“As well as assortativity-based metrics, the variance in node centrality provides an informative and intuitive network measure regarding the uniformity of a network’s structure, its resilience to perturbations, and the influence of start-points on social contagions (Freeman 1978; Borgatti 2005; Borgatti et al. 2006). For this purpose, we used node eccentricity (igraph package (Csardi & Nepusz 2006)), which measures how far a fisher is from the furthest other in the network (Hage & Harary 1995). Node eccentricity can be particularly informative when investigating the flow of information and transmission of behaviors across a network following an intervention (Table 2).”

We would like to highlight that we also explored variance in betweenness as an alternative to node eccentricity, and we presented these results in the Supplementary Materials. The results demonstrate that the two measures perform similarly.

Other than this, in Table 2 they authors cite ecological studies using the measure. In sum, I would expect a much better empirical, theoretical, support for their choice of metric.

We chose to include a non-human animal network example for degree assortativity in Table 2 as we believed it provided a clear and suitable example for demonstrating how assessing degree assortativity in networks can inform researchers. We are not aware of any small-scale fishery network studies that have specifically used degree assortativity (or degree homophily) as one of their key network measures. As the Reviewer highlights, the majority of small-scale fisher network studies have employed other types of homophily metrics based on actor attributes. However, we aimed to first understand the general structure of multiple fishing-related information types that relate to the sea turtle bycatch intervention in our study system and how each information-sharing network relates to one another. We believe that degree assortativity is a suitable network metric for this aim. Exploration of how actor attributes drive network centrality and grouping in each of the information-sharing networks is an excellent idea for further research and this study is underway. But we believe this analysis is substantive enough to warrant publication in an additional manuscript.

In light of these justifications, we acknowledge that including an ecological non-human animal network study as a key example of degree assortativity use may have been confusing to some readers. We have replaced the former example in Table 2 with a study that investigates degree assortativity (among other metrics) in a human social network:

Degree assortativity

A comprehensive, socio-centric network study of the Hadza hunter-gatherers of Tanzania was undertaken. Hadza networks were positively assorted by degree. People with higher in-degree named more social contacts, and people with higher out-degree were more likely to be named, even in models with controls. In other words, individuals who nominate more friends are more popular even among those they themselves did not nominate (Apicella et al. 2012).

Node eccentricity

A human social network example was already provided for node eccentricity.

The second major concern I have is that I am not sure that the authors' conclusions can be justified by their results. Specifically, the authors conclude that information-sharing networks about turtle bycatch are "structurally dissimilar" from the other information-sharing networks and, even more troubling—they conclude that the "usual mechanisms that drive information sharing between fishers in the other fishing-information types assessed (and potentially social networks generally) are not at play in the turtle bycatch information-sharing network". The problem I have with this conclusion is that degree assortativity is only one of many mechanisms that could be driving the formation of these information-sharing networks, and unfortunately, with the approach employed, there is no way that the authors can ascertain what are the mechanisms driving network formation (more on this below). My initial reaction was that in order to address this, the authors could refocus their conclusions to talk specifically about the observed differences in degree assortativity (rather than structure in general), but even then, the problem is that some, even more basic mechanisms of network formation (that are not

accounted for in the study) could actually explain the differences observed in degree assortativity. For example, reciprocity is commonly observed in information-sharing networks, and from what I can see in Figure 4 there seems to be a fair amount of reciprocity in the data. However, the authors do not mention these and other potential mechanisms, how they have accounted for them or why they are not relevant to their analysis or context (for example, homophily based on demographic characteristics such as race or family ties/clans, have been found as key mechanisms of network formation in fishers information-sharing networks). In other words, I think the authors have to do a much better job of supporting their methodological approach. If they cannot fully support their choice (and right now I don't think the support required is there), I would invite the authors to re-run their results using appropriate methods for the structural analysis of networks where multiple competing mechanisms can be tested concurrently.

Thanks for highlighting your concerns with our concluding statement. The Reviewer is correct that stating "structurally dissimilar" was perhaps an overstretch as unknown processes that have not been evaluated in the current study could also potentially be driving the formation of these information sharing network. We now offer further specificity by referring to the sea turtle bycatch network as "structurally dissimilar in degree assortativity and node eccentricity" to the other networks, rather than referring to the networks as structurally dissimilar more generally.

The overall aim of this study was to broadly compare the general network structure of multiple subtopics of fishing information that relate to a conservation intervention being implemented in our study system. We wanted to explore how the structures of information-sharing networks vary in relation to the topic of information shared and understand whether it would be possible to design a robust (if sub-optimal) intervention if influential individuals identified from an information-sharing network of relevance to the intervention other than sea turtle bycatch were targeted. To do this, we chose several widely applied network-level metrics that can be particularly informative about how networks structure varies and how that might influence information flows. We intended to explore whether individuals sharing one topic of information may not be the most central individuals when sharing other closely related information topics – allowing us to highlight to other conservation practitioners and researchers that there is a need to be careful when choosing the topic of information that one is mapping and to consider how it relates to the conservation intervention being planned. Essentially, we would like this study to be a timely word of warning as social network analysis is more widely applied in conservation science and natural resource management (Groce et al. 2019). We have seen more than one study in the conservation science literature that has mapped general 'fishing information' networks and then drawn more specific conclusions for conservation outcomes from that data (see Barnes et al. 2016, PNAS; Mbaru and Barnes 2017, Biological Conservation).

Furthermore, we agree with the Reviewer that other mechanisms may also be driving information sharing about fishing in our study systems networks. But we argue that the first step was to explore the general social structure of multiple subtopics of fishing information networks using three widely applied and generalisable network-level

metrics. Having found that the sea turtle bycatch information-sharing network is structurally different to the other networks in regards to these structural network measures, we are now carrying out further exploration into how demographic characteristics might be driving these differences. We have added three sentences in the discussion that highlight this point. The additional text reads:

“Additionally, other mechanisms could be expected to drive information sharing between fishers, prompting further research to investigate whether additional demographic characteristics correspond with different types of fishing information exchange. For example, several studies of small-scale fisheries have shown that similarity in gear type coincides with information exchange amongst fishers (Crona & Bodin 2006; Alexander et al. 2018). Similarly, longline fishers in Hawaii show a strong homophilic tendency for information exchange along ethnic lines (Barnes-Mauthe et al. 2013).”

Regarding the Reviewer’s comment on reciprocity. When considering the variance in betweenness as an alternative, but related, measure of centrality, we found it was positively correlated with eccentricity but negatively correlated with the clustering coefficient (Fig. S3; Supplementary Materials). The clustering coefficient, like reciprocity, is a quantitative measure used to study complex networks and can inform how much information is lost when a directed network is regarded as undirected. This was explored as an alternative measure to reciprocity during our analysis. This point is highlighted in the text of the Supplementary Materials.

Related to my last point above, I note that a previous reviewer highlighted ERGMs as a more stringent null model that is able to take into account multiple network-formation effects. In my opinion, the authors’ response to this major concern (as highlighted by the reviewer) is not convincing. The authors argue that while they acknowledge that ERGMs can perform many of the functions that their permutation null model approach can, they think there is still novelty in applying the permutation-based null model because they are only aware of 2 studies on human networks that use the same permutation null model that they use. This is not convincing for two reasons. First, there are tons of permutation-based models that have been used in the analysis of human networks since at least the 80s. They all vary slightly in what the null models are conditioned on, but they are all doing the same thing i.e. trying to get a test statistic for the measure of interest by comparing it to a baseline. So saying that “there is novelty in applying these analytical methods” is not enough.

We note in the text of the methodology section that numerous permutation-based null models have been applied to the analysis of human networks since the 1970s:

“Null models have been applied to network data in sociology since the 1970s (Davis 1970) and discipline-specific developments have subsequently been made to statistical models such as exponential random graph models (Snijders 2002; Snijders et al. 2006), conditional uniform graph tests (Katz & Wilson 1956; Holland & Leinhardt 1977; Anderson et al. 1999) and quadratic assignment procedure tests (Hubert 1986; Krackardt 1987; Dekker et al. 2007).”

While we do not explicitly state there is novelty in our application of our specific permutation methodology, we do note that we expand on the application of these specific permutation analyses beyond the original field of application. We feel that, should an analytical method have application beyond a field of study with which it was first designed and routinely applied, then there is benefit in demonstrating its flexibility and applicability to other areas of research. Therefore, we have kept the following sentence in the methods to highlight this point:

“Here we expand the application of the permutation-based null model approach routinely used in ecology, and which has also been applied in the field of epidemiology for assessing human contact tracing disease control measures, to an human information-sharing social network (Firth et al. 2020).”

But we have removed the following text from the final paragraph of the introduction to deemphasise the novelty of applying this permutation method to human network data:

“We illustrate how null model analysis techniques used in the ecological sciences may offer more in-depth insights into the fine-scale structure of human networks than could be gained from simple centrality measurement methods”

Second, the point about ERGMs is not that they can perform many of the functions that permutations test can. The point is that these models can perform all of the functions that the permutation model applied by the authors can, plus many more. ERGMs are much more advanced models that seem to me are much more suitable for the task at hand. This not to say that permutation null models are not adequate for certain tasks. But given that the authors have not fully supported why it makes sense to ONLY compare the structure of the networks on the basis of their degree assortativity and, also importantly, why the particular permutations implemented are suitable given the particular context and type of data, I am afraid I am not convinced that the permutation null models selected are suitable.

Thanks for these comments. We agree with the Reviewer that ERGMs are a useful analytical technique for comparing social networks. As highlighted in our comment above we have highlighted the use of ERGMs and other null model approaches routinely applied in the sociology literature when comparing the structure of social networks. However, in the current study we have chosen to compare multiple human information-sharing networks using a permutation method that has historically been routinely applied to non-human animal social networks. The permutation null models that we have applied can also perform many more functions beyond what are demonstrated in this study. Both the permutation null models that we have applied and ERGMs are highly flexible, and we are not arguing for one over the other. So, unfortunately, we do not necessarily agree with the Reviewer’s statement “*The point is that these models [ERGMs] can perform all of the functions that the permutation model applied by the authors can, plus many more*”. While our approach may be different from the analytical methods usually applied to comparing human social networks, there is no

reason to preclude the use of the permutation null models that we have applied to the human network dataset that we are exploring. The permutation method we employ is suitable for the task of comparing whether certain network metrics are more or less expected than by chance across multiple information-sharing networks. Our graph permutation techniques are robust, and the method has also successfully been applied to human social network data previously - see (Firth et al. 2020). Furthermore, Reviewer 1 notes “*the permutation exercise that we employ is interesting but lacks the nuances of more ‘constraining’ null models that maintain, for example, the number of triangles in the network [like graphlets as described in Przuji (2007) and Tantardini et al. (2019)]. Still it can work as a coarse grained assessment of whether certain network metrics are more or less expected than by chance*”.

In an effort to provide further clarification and justification for the structural comparison of networks undertaken we have also added a paragraph in the introduction to make a stronger case as to why the assumption, made by conservation researchers and practitioners, that knowledge about peer-to-peer information-sharing networks should be transferable to a related information-sharing network of interest (other fishing issues and sea turtle bycatch, in our case) should be tested:

“A structural comparison of multiple fishing information-sharing networks will allow us to explore whether it would be possible to design an effective (if sub-optimal) sea turtle bycatch intervention by identifying and targeting influential individuals in a network sharing information on other topics related to the intervention. This is a pertinent question because information-sharing about sea turtle bycatch might be sensitive and therefore hard to quantify, or it may be that the information-sharing networks for other topics are already known so the cost of describing a sea turtle bycatch-relevant network would not need to be incurred. It is also interesting more generally to explore how the networks for sharing different types of fisheries information resemble each other, as this may give insights into how different kinds of information are perceived by fishers.”

I provide minor comments in the attached pdf document.

We have provided detailed responses to the Reviewer’s minor comments in the attached .pdf document.

Finally, we thank the Reviewer for their thorough and constructive critique of our research.

Literature cited

- Alexander S, Bodin Ö, Barnes M. 2018. Untangling the drivers of community cohesion in small-scale fisheries. *International Journal of the Commons* **12**:519–547.
- Anderson BS, Butts C, Carley K. 1999. The interaction of size and density with graph-level indices. *Social networks* **21**:239-267.
- Apicella CL, Marlowe FW, Fowler JH, Christakis NA. 2012. Social networks and cooperation in hunter-gatherers. *Nature* **481**:497.

- Barnes ML, Lynham J, Kalberg K, Leung P. 2016. Social networks and environmental outcomes. *Proceedings of the National Academy of Sciences* **113**:6466-6471.
- Barnes-Mauthe M, Arita S, Allen S, Gray S, Leung P. 2013. The influence of ethnic diversity on social network structure in a common-pool resource system: implications for collaborative management. *Ecology and Society* **18**.
- Borgatti SP. 2005. Centrality and network flow. *Social networks* **27**:55-71.
- Borgatti SP, Carley KM, Krackhardt D. 2006. On the robustness of centrality measures under conditions of imperfect data. *Social networks* **28**:124-136.
- Centola D. 2011. An Experimental Study of Homophily in the Adoption of Health Behavior. *Science* **334**:1269-1272.
- Centola D 2018. *How Behavior Spreads: The Science of Complex Contagions*. Princeton University Press.
- Crona B, Bodin Ö. 2006. What you know is who you know? Communication patterns among resource users as a prerequisite for co-management. *Ecology and society* **11**.
- Csardi G, Nepusz T. 2006. The igraph software package for complex network research. *InterJournal, Complex Systems* **1695**:1-9.
- Davis JA. 1970. Clustering and hierarchy in interpersonal relations: Testing two graph theoretical models on 742 sociomatrices. *American Sociological Review*:843-851.
- Dekker D, Krackhardt D, Snijders TA. 2007. Sensitivity of MRQAP tests to collinearity and autocorrelation conditions. *Psychometrika* **72**:563-581.
- Firth JA, et al. 2020. Using a real-world network to model localized COVID-19 control strategies. *Nature Medicine* **26**:1616-1622.
- Freeman LC. 1978. Centrality in social networks conceptual clarification. *Social networks* **1**:215-239.
- Groce JE, Farrelly MA, Jorgensen BS, Cook CN. 2019. Using social-network research to improve outcomes in natural resource management. *Conservation Biology* **33**:53-65.
- Hage P, Harary F. 1995. Eccentricity and centrality in networks. *Social networks* **17**:57-63.
- Holland PW, Leinhardt S. 1977. A method for detecting structure in sociometric data. Pages 411-432. *Social Networks*. Elsevier.
- Hubert L 1986. *Assignment methods in combinational data analysis*. CRC Press.
- Katz L, Wilson TR. 1956. The variance of the number of mutual choices in sociometry. *Psychometrika* **21**:299-304.
- Krackardt D. 1987. QAP partialling as a test of spuriousness. *Social networks* **9**:171-186.
- Mbaru EK, Barnes ML. 2017. Key players in conservation diffusion: Using social network analysis to identify critical injection points. *Biological Conservation* **210**:222-232.
- McPherson M, Smith-Lovin L, Cook JM. 2001. Birds of a Feather: Homophily in Social Networks. *Annual Review of Sociology* **27**:415-444.
- Newman ME. 2002. Assortative mixing in networks. *Physical review letters* **89**:208701.
- Newman ME. 2003. Mixing patterns in networks. *Physical Review E* **67**:026126.

- Newman ME, Park JJPrE. 2003. Why social networks are different from other types of networks. **68**:036122.
- Snijders TA. 2002. Markov chain Monte Carlo estimation of exponential random graph models. *Journal of Social Structure* **3**:1-40.
- Snijders TA, Pattison PE, Robins GL, Handcock MS. 2006. New specifications for exponential random graph models. *J Sociological methodology* **36**:99-153.
- Weiss K, Hamann M, Kinney M, Marsh H. 2012. Knowledge exchange and policy influence in a marine resource governance network. *Global Environmental Change* **22**:178-188.
- Zhang AJ, Matous P, Tan DKY. 2020. Forget opinion leaders: the role of social network brokers in the adoption of innovative farming practices in North-western Cambodia. *International Journal of Agricultural Sustainability*:1-19.